# Metacognition across domains: Is the association between arithmetic and metacognitive monitoring domain-specific?

**Elien Bellon**[1]*, **Wim Fias**[2], **Bert De Smedt**[1]

**1** Parenting and Special Education Research Unit, KU Leuven, Leuven, Belgium, **2** Experimental Psychology, Ghent University, Gent, Belgium

* elien.bellon@kuleuven.be

**Data Availability Statement:** An anonymized dataset will be available on the Open Science Framework (https://osf.io/ypue4/?view_only=

## Abstract

Metacognitive monitoring is a critical predictor of arithmetic in primary school. One outstanding question is whether this metacognitive monitoring is domain-specific or whether it reflects a more general performance monitoring process. To answer this conundrum, we investigated metacognitive monitoring in two related, yet distinct academic domains: arithmetic and spelling. This allowed us to investigate whether monitoring in one domain correlated with monitoring in the other domain, and whether monitoring in one domain was predictive of performance in the other, and vice versa. Participants were 147 typically developing 8-9-year-old children (Study 1) and 77 typically developing 7-8-year-old children (Study 2), who were in the middle of an important developmental period for both metacognitive monitoring and academic skills. Pre-registered analyses revealed that within-domain metacognitive monitoring was an important predictor of arithmetic and spelling at both ages. In 8-9-year-olds the metacognitive monitoring measures in different academic domains were predictive of each other, even after taking into account academic performance in these domains. Monitoring in arithmetic was an important predictor of spelling performance, even when arithmetic performance was controlled for. Likewise, monitoring in spelling was an important predictor of arithmetic performance, even when spelling performance was controlled for. In 7-8-year-olds metacognitive monitoring was domain-specific, with neither correlations between the monitoring measures, nor correlations between monitoring in one domain and performance in the other. Taken together, these findings indicate that more domain-general metacognitive monitoring processes emerge over the ages from 7 to 9.

## Introduction

"Learn from your mistakes" is an old saying that (grand)parents teach their children. This goes back to the premise that making mistakes is associated with learning. Noticing your mistakes is an example of monitoring your cognition. This monitoring of cognition is a facet of metacognition, a concept first introduced by Flavell [1]. One critical component of metacognition is procedural metacognition. This is a self-reflecting, higher-order cognitive process, which indicates how people monitor and control their cognition during ongoing cognitive processes [2,3]. Metacognitive monitoring is an important aspect of procedural metacognition and is

ce9f97af0e3149c28942a43499eafd32) after publication.

**Funding:** This research was supported by the Fund for Scientific Research Flanders grant G.0638.17 The funders had no role in study design, data collection and analysis, decision to publish or preparation of the manuscript.

**Competing interests:** The authors have declared that no competing interests exist.

defined as the subjective self-assessment of how well a cognitive task will be/is/has been performed [3,4].

Two recent studies found evidence for metacognitive monitoring (i.e., reflecting procedural metacognition) as an important predictor of arithmetic achievement [5,6]. To determine the role of metacognitive monitoring, these authors asked children on a trial-by-trial basis to report their judgement of the accuracy of their answers during an arithmetic task. Both studies found that successful appraisal of the accuracy of one's arithmetic judgement is a powerful predictor of arithmetic performance in primary school children. To date, however, it is unclear whether the results regarding the strength of the role of metacognitive monitoring in arithmetic are specific to the arithmetic domain, or whether they are reflective of a more general role of metacognitive monitoring in academic performance; an outstanding question on which this study will focus.

Metacognition has been regarded as a fundamental skill influencing cognitive performance and learning in domains as diverse as arithmetic, memory, reading, perception, and many others (e.g., [6–18]). The importance of metacognition that was found in existing research in different (cognitive) domains is not surprising, as metacognitive aspects, such as knowing the limits of your own knowledge and being able to regulate that knowledge, are essential components of self-regulated and successful learning [16], enabling learners to improve their cognitive achievements. For example, good metacognition allows learners to correctly allocate study-time, check answers when they feel unsure about the correctness of the answer or provide a learning moment when an error is detected. Besides being considered a global ability playing a role in a large range of domains, metacognition, and consequently metacognitive monitoring, is usually considered to be a domain-general cognitive process that is correlated across content domains. This suggests that people who are good at evaluating their performance for one task, also tend to be good at evaluating their performance for another task (e.g., [19,20]). There is, however, evidence suggesting that this domain-generality only emerges over development. Geurten and colleagues [19] recently observed that metacognition is first domain-specific and then generalizes across domains as children mature. They found a gradual shift from domain-specific towards domain-general metacognition across the arithmetic and memory domains in children aged between 8 and 13. In adults, more evidence for the domain-generality has been observed. Veenman and colleagues [21] and Schraw and colleagues [20,22] found that metacognitive measures are correlated across unrelated (cognitive) tasks. More specifically, Schraw and colleagues [20] found significant correlations between metacognitive measures across eight different domains ranging from historic knowledge to knowledge of caloric values of food. This domain-general hypothesis in adults is also supported by brain imaging data that show that adults' metacognitive abilities for different types of tasks partially depend on common neural structures, such as the prefrontal cortex [23] and precuneus [24].

However, domain-specific knowledge and skills also seem to be important for metacognitive monitoring. For example, in young children (ages 5 to 8 years), Vo and colleagues [25] showed that metacognition in the numerical domain was unrelated to metacognition in the emotional domain, suggesting young children's metacognition is domain-specific. Based on their empirical findings, Schraw and colleagues [20] suggested that in adults metacognitive monitoring within a specific domain is governed by general metacognitive processes in addition to domain-specific knowledge. Löffler, Von Der Linden and Schneider [26] documented a twofold effect of expertise on monitoring in soccer: Although domain-specific knowledge enhances monitoring performance in some situations, more optimistic estimates (presumably due to the application of a familiarity heuristic) typically reduce monitoring accuracy in experts. Likewise, in mathematics, metacognitive monitoring has been found to be a function of domain-specific ability (e.g., [27,28]). Taken together, the existing research also illustrates the importance of domain-specific knowledge and skills for metacognitive monitoring.

This issue of domain-specificity is a longstanding debate within the metacognitive literature (e.g., [19–22,29]), both on the behavioural and brain-imaging level. Yet, in children, the results are scarce and rather inconclusive, with different results for various age groups as well as metacognitive measures.

Firstly, age-related differences in the results on domain-specificity of metacognition in children are not surprising, as a critical development in monitoring is observed during early to late childhood (e.g., [19,30]). For example, in (early) primary school, metacognitive monitoring accuracy is found to increase (e.g., [30–35]). In the same developmental time period of these age-related improvements in monitoring of cognition, there are also important age-related improvements in academic skills, such as arithmetic and spelling. The age-related metacognitive improvements are recognized to underlie several aspects of cognitive development in various domains (e.g., improvements in accuracy; e.g., [30]). Furthermore, based on their empirical findings, Geurten and colleagues [19] conclude that a gradual shift toward domain-general metacognition occurs in children between 8 and 13 years, and that metacognition is no more bound by task content and domain knowledge after the age of 10. Against this background and to thoroughly investigate the domain-specificity question in children, the current research specifically recruited 8–9 year-olds (third grade; Study 1) and 7–8 year-olds (second grade; Study 2), who are in the middle of this important developmental period for both metacognitive monitoring and academic skills.

Secondly, the different results on domain-specificity of metacognition in children for different metacognitive measures may in part be due to different aspects of metacognition being investigated. Metacognition includes both declarative and procedural metacognition. As metacognition encompasses different aspects, it is not surprising that these different aspects of metacognition follow different developmental paths [34] and that they are differently associated with domain-specific knowledge and skills. A recent study by Bellon and colleagues [5], for example, found that within-domain metacognitive monitoring was associated with arithmetic performance, while declarative metacognitive knowledge was not. The authors suggest this might indicate that children's metacognition is more domain-specific than it is domain-general. Yet, the authors based their suggestion on results on different aspects of metacognition, which were measured fundamentally differently (i.e., online, trial-by-trial reports for metacognitive monitoring vs. general questionnaire for declarative metacognitive knowledge), making testing the domain-specificity hypothesis as well as making strong claims about domain-specificity of metacognition troublesome.

To overcome these issues, the current research specifically focused on the monitoring aspect of metacognition. Extending the existing body of data, we included, in addition to the metacognitive monitoring measure in arithmetic, the same metacognitive monitoring measure in another domain of academic learning, i.e., spelling. By including metacognitive monitoring measures in two domains, and, importantly, by using the exact same paradigm to measure it, the current study was able to investigate the question of domain-specificity more thoroughly. The paradigm to measure metacognitive monitoring was the same as in Bellon et al. [5] and Rinne and Mazzocco [6]. Spelling was included as a second domain to maximize the comparability of the two tasks in which metacognitive monitoring was measured. Arithmetic and spelling are quintessential domains in primary school and in both domains primary school children go through crucial developmental steps. Based on the children's curriculum, we were able to select age-appropriate items. This allowed us to thoroughly investigate whether the results on the role of metacognitive monitoring in arithmetic are specific to the arithmetic domain or not.

Based on the outstanding issues outlined above, this study aims to extend and deepen our knowledge on the domain-specificity of the role of metacognition in different academic

domains in middle childhood. Specifically, this study will investigate whether metacognitive monitoring is domain-specific or not by investigating (a) the associations between within-domain metacognitive monitoring and arithmetic and spelling; (b) whether metacognitive monitoring in one domain is associated with and/or predicted by metacognitive monitoring in the other domain; (c) whether performance in one domain is associated with and/or predicted by metacognitive monitoring in the other domain, and (d) these questions in two different age groups in primary school to fully grasp potential transitional periods in the domain-specificity of metacognitive monitoring.

If, on the one hand, metacognition is highly domain-general, then metacognitive monitoring in the arithmetic and spelling tasks will be correlated and predictive of each other, even when controlled for academic performance–as arithmetic and spelling are highly related domains; and metacognitive monitoring in one domain will be associated with and predicts academic performance in the other domain. If, on the other hand, metacognition is highly domain-specific, then the associations described above will be non-significant (frequentist statistics) and Bayes factors will be close to zero (Bayesian statistics; see below). These questions are investigated in two different age groups for which, based on the existing literature, different predictions can be made on the extent to which metacognitive monitoring is domain-general. By selecting participants in these two age groups, we aimed to capture an important period in the development of (domain-generality of) metacognitive monitoring. In Study 1, we investigated these questions in 8-9-year-olds, for which domain-generality of metacognitive monitoring was predicted (third grade). Study 2 investigated these questions in younger children, namely 7-8-year-olds, for which more domain-specificity of metacognitive monitoring was predicted (second grade).

## Study 1: Metacognitive monitoring in arithmetic and spelling in 8-9-year-olds (third grade)

### Methods

**Participants.** Participants were 147 typically developing Flemish 8–9 year-olds (third grade; 69 girls; $M_{age}$ = 8 years, 10 months; SD = 3 months; [8 years 4 months—9 years 4 months]), without a diagnosis of a developmental disorder, and who came from a dominantly middle-to-high socio-economic background. This study was approved by the social and societal ethics committee of KU Leuven. For every participant, written informed parental consent was obtained.

**Procedure.** All participants participated in four test sessions, which took place at their own school during regular school hours. They all completed the tasks in the same order. In the context of a larger project, all children first participated in an individual session of which the data are not included in the current manuscript. Second, a session in small groups of eight children took place, including the computerized spelling task and motor speed task. Third, a second session in small groups took place, including the computerized arithmetic task and motor speed tasks. Fourth, in a group session in the classroom, the standardized arithmetic and spelling tests and the test of intellectual ability were administered. Sessions were separated by one to three days on average; they were never adjacent. Below we describe the key variables and control variables used to answer our research questions. The full cognitive testing battery is posted on the Open Science Framework (OSF) page of this project (https://osf.io/ypue4/?view_only=ce9f97af0e3149c28942a43499eafd32).

**Materials.** Materials consisted of written standardized tests and computer tasks designed with Open Sesame [36]. Arithmetic and spelling skills were assessed with both a custom computerized task and a standardized test (i.e., Arithmetic: Tempo Test Arithmetic [37]; Spelling:

standardized dictation [38]). The computerized tasks for arithmetic and spelling were specifically designed to be as similar as possible, to minimize the possibility that the results on domain-specificity of metacognition were due to differences in paradigm. Both tasks were multiple choice tasks with specifically selected age-appropriate items (i.e., single digit addition and multiplication for arithmetic; three specific Dutch spelling rules for spelling). After a first introductory block, in the second block of each task, participants had to report their judgment on the accuracy of their academic answer after each trial, using the same metacognitive monitoring measure in both tasks.

**Arithmetic.** *Custom computerized arithmetic task*. This single-digit task included addition and multiplication items, and comprised all combinations of the numbers 2 to 9 ($n = 36$). The task consisted of two blocks, i.e., one introductory block without ($n = 12$) and one with ($n = 60$) a metacognitive monitoring measure (see below). Stimuli were pseudo-randomly divided into the two blocks and children were given a short break between blocks. Each block was preceded by four practice trials to familiarize the child with the task requirements. Performance on the practice items was not included in the performance measures. In both blocks, addition items were presented first ($n = 6$ in the first block; $n = 30$ in the second block). After a short instruction slide indicating multiplication items would follow, the multiplication items were presented ($n = 6$ in the first block; $n = 30$ in the second block). The position of the numerically largest operand was balanced. Each item was presented with two possible solutions, one on the left and one on the right side of the screen. In half of the items, the correct solution was presented on the left side of the screen. Incorrect solutions for the addition items were created by adding or subtracting 1 or 2 to the solution ($n = 7$ for every category), or by using the answer to the corresponding multiplication item (e.g., 6 + 3 with incorrect solution 18; $n = 8$). The incorrect solutions for the multiplication items were table related, i.e., solution -1 or +1 one of the operands (e.g., 6 × 3 = 24; $n = 7$ for every category), or the answer to the corresponding addition (e.g., 8 × 2 = 10; $n = 8$). Each trial started with a 250 ms fixation point in the centre of the screen and after 750 ms the stimulus appeared in white on a black background. The stimuli remained visible until response. The children had to indicate which of the presented solutions for the problem was correct (by pressing the corresponding key). The response time and answer were registered via the computer. Performance measures were both accuracy and the response time for correct answers in the second block ($n = 60$).

*Standardized arithmetic task*. Arithmetic fluency was assessed by the Tempo Test Arithmetic (TTA; [37]); a standardized pen-and-paper test of arithmetical fluency, which comprises five columns of arithmetic items (one column per operation and a mixed column), each increasing in difficulty. Participants got one minute per column to provide as many correct answers as possible. The performance measure was the total number of correctly solved items within the given time (i.e., total score over the five columns).

**Spelling.** *Custom computerized spelling task*. Spelling performance was measured with a computerized task consisting of two blocks, i.e., one introductory block without ($n = 12$) and one with ($n = 60$) a metacognitive monitoring measure (see below). Stimuli were pseudo-randomly divided into the two blocks and children were given a short break between blocks. Each block was preceded by six practice trials to familiarize the child with the task requirements. Performance on the practice items was not included in the performance measures. The items consisted of a Dutch word with a missing part, that was replaced by an underscore (e.g., 'ko_ie' for 'koffie'), presented with two possible solutions, one on the left and one on the right side of the screen. We used three specific Dutch spelling rules, which were the focus of spelling instruction at the participants' age. Firstly, the rule of open and closed syllables was used, on the basis of which one can figure out if one or two vowels or consonants have to be written.

Secondly, the extension rule was used, on the basis of which one can figure out if words with a [t] sound at the end of the word are written with a 't' or a 'd'. To correctly spell these two types of words, children can either use these rules, or when they have extensive experience with these words, retrieve the correct spelling from their memory. Flemish third graders have the most experience with the extension rule, and are in the learning phase for the open and closed syllables rule. Stepwise, they go from learning the rule and using the procedure to spell the words, towards automatization of the correct spelling and thus retrieving it from memory. This spelling development is analogous to arithmetic development in third grade (i.e., from procedure use to retrieval). A third category of words was added for which no rule is available, but only retrieval from long-term memory is possible (i.e., au/ou-words; ei/ij-words). The diphthongs in these words have the same pronunciation, but are spelt differently (e.g., 'r*ei*s' vs. 'w*ij*s' have both the [ɛi] sound)–there is no rule to determine whether one or the other diphthong should be used and children have to learn this by heart. All items were selected from curriculum-based glossaries. Incorrect solutions were created by using the related distractor (*n* = 14 for each category), namely one or two vowels or consonants for the open and closed syllables rule (e.g., koffie: 'ko_ie' with options 'f' or 'ff'), 't' or 'd' for the extension rule (e.g., kast: 'kas_' with options 't' or 'd'), and the related diphthong for the to-be-retrieved words (e.g., konijn: 'kon_n' with options 'ei' or 'ij'). In half of the items, the correct solution was presented on the left side of the screen. Each trial started with a 250 ms fixation point in the centre of the screen and after 750 ms children were presented on audiotape with the word. Then, the visual stimulus appeared in white on a black background. The stimuli remained visible until response. The children had to indicate which of the presented solutions for the problem was correct (by pressing the corresponding key; i.e., left/right key). The response time and answer were registered via the computer. Performance measures were both accuracy and the response time for correct answers in the second block (*n* = 60).

*Standardized spelling task*. Spelling ability was also measured with a standardized dictation [38]. We administered the subtest for children at the end of third grade, which includes age-appropriate, curriculum-based items. The experimenter read aloud 43 sentences and the participants had to write one word down that was repeated two times after the sentence was read. The performance measure was the total number of correctly written words.

**Metacognitive monitoring.**   In the second block of the arithmetic and the spelling task (*n* = 60 for each task), a metacognitive monitoring measure was added to the items. Children had to report their judgment on the accuracy of their answer to the item on a trial-by-trial basis (e.g., [5,6]). After giving their answer to the arithmetic/spelling problem, children had to indicate if they thought their answer was *Correct*, *Incorrect*, or if they *Did not know*. We used emoticons in combination with the options (e.g., ☺ and *Correct*) to make the task more attractive and feasible for children (Fig 1). Children had to respond by pressing the key corresponding to their metacognitive judgment (i.e., indicated with corresponding emoticon stickers). Metacognitive monitoring skills were operationalised as calibration of this judgment (i.e., the alignment between one's judgment in the accuracy of their answer to a problem and the actual accuracy of the answer). Namely, a calibration score of 2 was obtained if their metacognitive judgment corresponded to their actual performance (i.e., metacognitively judged as *Correct* and indeed correct academic answer; metacognitively judged as *Incorrect* and indeed incorrect academic answer), a calibration score of 0 if their metacognitive judgement did not correspond to their actual performance (i.e., metacognitively judged as *Correct* and in fact incorrect academic answer; metacognitively judged as *Incorrect* and in fact correct academic answer), and a calibration score of 1 if children indicated they *Did not know* about their academic answer. The metacognitive monitoring score per child was the mean of all calibration scores (i.e., calibration score per arithmetic/spelling item; *n* = 60 per domain) and was calculated for each task

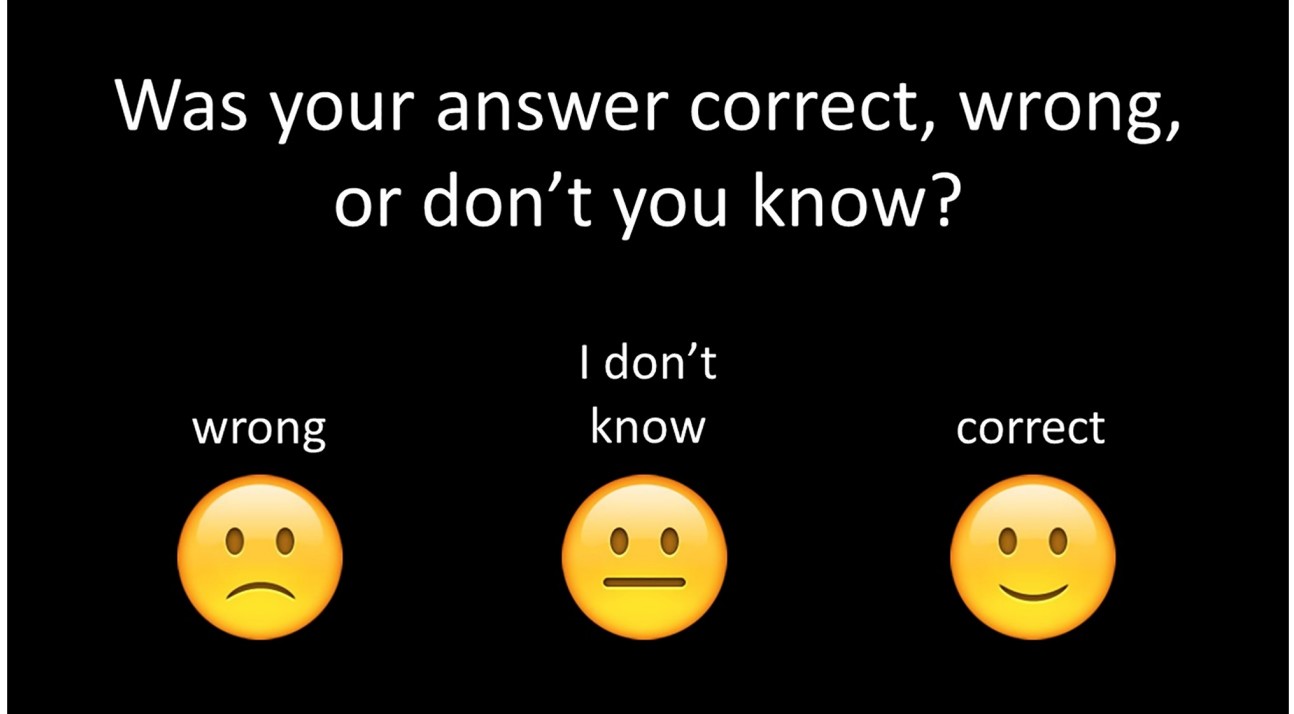

**Fig 1. Example of metacognitive monitoring question after arithmetic/spelling item.**

separately. The higher the calibration scores, the better the metacognitive monitoring skills. To familiarize the children with the task, practice items were presented in each task.

**Control variables.** *Intellectual ability*. Intellectual ability was assessed through the Raven's Standard Progressive Matrices [39]. Children were given 60 multiple-choice items in which they had to complete a pattern. The performance measure was the number of correctly solved patterns.

*Motor speed*. A motor speed task was included as a control for children's response speed on the keyboard [5]. Two shapes were simultaneously presented on either side of the screen and children had to indicate which of the two shapes was filled by pressing the corresponding key (i.e., left/right key). All shapes were similar in size and each shape occurred four times filled and four times non-filled, yielding 20 trials. The position of the filled shape was balanced. After fixation, stimuli appeared until response. Three practice trials were included to familiarize the children with the task. The performance measure was the average response time of correct responses.

**Data analysis.** A comprehensive analyses plan was preregistered on the OSF page of this project (https://osf.io/ypue4/?view_only=ce9f97af0e3149c28942a43499eafd32). The key analyses to answer our research questions are presented below; the results of the remaining preregistered analyses can be found in the supplementary materials (S1 File).

We ran frequentist analyses using both uni- and multivariate techniques, as well as Bayesian analyses. Frequentist analyses allowed us to explore our data by means of a well-known method to gauge statistical support for the hypotheses of interest. Bayesian statistics allowed us to test the degree of support for a hypothesis (i.e., degree of strength of evidence in favour of or against any given hypothesis), expressed as the Bayes factor (BF; the ratio between the evidence in support of the alternative hypothesis over the null hypothesis ($BF_{10}$)). Although Bayes factors

provide a continuous measure of degree of evidence, there are some conventional approximate guidelines for interpretation ([40] for a classification scheme): $BF_{10} = 1$ provides no evidence either way, $BF_{10} > 1$ anecdotal, $BF_{10} > 3$ moderate, $BF_{10} > 10$ strong, $BF_{10} > 30$ very strong and $BF_{10} > 100$ decisive evidence for the alternative hypothesis; $BF_{10} < 1$ anecdotal, $BF_{10} < 0.33$ moderate, $BF_{10} < 0.10$ strong, $BF_{10} < 0.03$ very strong and $BF_{10} < 0.01$ decisive evidence for the null hypothesis. By adding these Bayesian analyses, we deepened our findings from the traditional analyses, as we were able to identify evidence in favour of the null hypothesis, consequently, identify which hypothesis is most plausible (i.e., alternative hypothesis vs. null hypothesis) and which predictors are the strongest. This is particularly relevant for the current study because we can compare the strength of evidence in favour of the domain-general hypothesis (i.e., association between metacognitive monitoring measures in different domains; association between performance and metacognitive monitoring across domains) versus the domain-specific hypothesis (i.e., no association between metacognitive monitoring measures in different domains; no association between performance and metacognitive monitoring across domains).

To answer our research questions, we used correlation and regression analyses. For the Bayesian analyses, we used a default prior with prior width set to 1 for Pearson correlations and to .354 for the linear regression analyses. For Bayesian regressions, a $BF_{inclusion}$ was calculated for every predictor in the model, which represents the change from prior to posterior odds (i.e., $BF_{10}$), where the odds concern all the models with a predictor of interest to all models without that predictor (i.e., a Bayes factor for including a predictor averaged across the models under consideration).

As planned in the preregistration, we excluded a child's performance on a task if this performance was more than three standard deviations from the mean of the task (i.e., $\leq 3\%$ of the data per task). Due to unforeseen circumstances during data collection (e.g., school bell ringing), we additionally excluded some data at the item level (i.e., $< 0.57\%$ of items per task) that were considered to be measurement errors, i.e., when the data point was an outlier (i.e., more than three standard deviations from the mean) on both the item level (i.e., compared to the general mean of the item) and on the subject level (i.e., compared to the personal mean of the subject).

## Results

The descriptive statistics of all measures are presented in S1 Appendix. Additionally, Pearson correlation coefficients of all variables under study were calculated (S2 Appendix). Although not originally pre-registered, we additionally re-calculated all analyses below with chronological age as an additional control variable. Considering chronological age within grade in the analyses reported below did not change the interpretation of the results (S3 Appendix).

**The role of metacognitive monitoring in arithmetic and spelling performance.** Pearson correlation coefficients of the associations between metacognitive monitoring and the academic skills are presented in Table 1.

Metacognitive monitoring in the arithmetic task ($MM_{arith}$) was significantly correlated with arithmetic accuracy ($Arithmetic_{acc}$) and the tempo test arithmetic (TTA), with Bayes factors indicating decisive evidence in favour of the associations, even when controlling for intellectual ability. There was no significant correlation with response time for correct arithmetic answers ($Arithmetic_{rt}$) and the Bayes factor indicated moderate evidence in favour of no association.

Metacognitive monitoring in the spelling task ($MM_{spell}$) was significantly correlated with spelling accuracy ($Spelling_{acc}$) and dictation, with Bayes factors indicating decisive evidence in favour of the associations, even when controlling for intellectual ability. There was no

**Table 1. Correlation analyses of metacognitive monitoring and academic performance measures in 8-9-year-olds (Grade 3).**

| | Arithmetic | | | Spelling | | |
|---|---|---|---|---|---|---|
| | Custom task–Accuracy[a] | Custom task—RT[b] | Standardized task (TTA)[a] | Custom task—Accuracy[a] | Custom task -RT[b] | Standardized task (dictation)[a] |
| Metacognitive Monitoring | | | | | | |
| *Arithmetic* | | | | | | |
| r | .84 | -.08 | .38 | .45 | .11 | .26 |
| p | <.001 | .38 | <.001 | <.001 | .20 | .003 |
| BF$_{10}$ | >100 | 0.16 | >100 | >100 | 0.24 | 9.65 |
| *Spelling* | | | | | | |
| r | .48 | -.19 | .33 | .91 | -.02 | .66 |
| p | <.001 | 0.03 | <.001 | <.001 | .79 | <.001 |
| BF$_{10}$ | >100 | 1.18 | >100 | >100 | 0.11 | >100 |

[a] Controlled for intellectual ability.

[b] Controlled for intellectual ability and motor speed on the keyboard.

significant correlation with response time for correct spelling answers (Spelling$_{rt}$) and the Bayes factor indicated moderate evidence in favour of no association.

Based on the absence of significant (frequentist statistics) and supported (Bayesian statistics) associations with our response time performance measures (Arithmetic$_{rt}$ and Spelling$_{rt}$), and because these measures only take into account data for correct answers, losing important information on performance and possibly overestimating performance, the response time performance measures will not be considered in further analyses.

**Domain-specificity of the role of metacognitive monitoring.** To examine domain-specificity of the role of metacognition, we first investigated the association between MM$_{arith}$ and MM$_{spell}$ with correlation and regression analyses. Specifically, we investigated whether MM$_{arith}$ and MM$_{spell}$ were correlated, even when controlling for intellectual ability and academic performance in both domains. Controlling for intellectual ability and performance in both standardized academic tasks was necessary, to make sure the observed associations between MM$_{arith}$ and MM$_{spell}$ were not (entirely) driven by their shared reliance on intellectual ability or by the high correlation between both academic domains.

Secondly, we studied the role of MM$_{spell}$ in arithmetic performance and MM$_{arith}$ in spelling performance with correlation and regression analyses. In other words, cross-domain correlations between academic performance in one domain and metacognitive monitoring in the other domain were calculated. As performance in the arithmetic and spelling tasks was highly correlated, the cross-domain associations of metacognitive monitoring and academic performance might rely on the correlation between the academic tasks. Therefore, we used regression models to investigate whether metacognitive monitoring in arithmetic uniquely predicted spelling performance on top of arithmetic performance, and vice versa.

In a final step, we investigated the unique contribution of cross-domain metacognitive monitoring to performance over within-domain metacognitive monitoring using regression models including metacognitive monitoring in both domains as predictors for academic performance.

*Associations between metacognitive monitoring in different domains.* MM$_{arith}$ and MM$_{spell}$ were significantly correlated, even when controlling for intellectual ability, and arithmetic and spelling performance on the standardized tasks ($r = .42$; $p < .001$; BF$_{10} > 100$). Regression analyses confirmed that metacognitive monitoring in one domain was uniquely predicted by

metacognitive monitoring in the other domain, even when simultaneously considered with intellectual ability and performance on the standardized tasks in both academic domains (see Table 2). Additional post-hoc analyses that were not preregistered indicated that the results were the same when including academic achievement as measured with accuracy in the computerized academic tasks instead of academic achievement as measured with the standardized academic tasks.

*Cross-domain performance associations of metacognitive monitoring.* Table 1 shows cross-domain correlations between academic performance and metacognitive monitoring in the other domain. $MM_{arith}$ was significantly correlated with both spelling performance measures (i.e., $Spelling_{acc}$ and dictation), with a Bayes factor indicating moderate to decisive evidence. $MM_{spell}$ was significantly correlated with both arithmetic performance measures (i.e., $Arithmetic_{acc}$ and TTA), with a Bayes factor indicating decisive evidence.

We further investigated whether metacognitive monitoring in arithmetic uniquely predicted spelling performance on top of arithmetic performance; and vice versa. Namely, we predicted arithmetic performance based on $MM_{spell}$ and dictation, and spelling performance based on $MM_{arith}$ and TTA (Table 3). These regression analyses showed that, even when performance in the academic domain was taken into account, metacognitive monitoring in that domain remained a significant and supported predictor of academic performance in the other domain (all $p$s $< .05$; all BFs$_{10}$ $>5$).

When metacognitive monitoring scores in both domains were considered simultaneously to predict academic performance (using regression analyses), only the role of metacognitive monitoring within the domain itself remained significant (frequentist statistics) and supported (Bayesian statistics). Namely, when $MM_{arith}$ and $MM_{spell}$ were used to predict arithmetic performance, only $MM_{arith}$ was a significant and supported predictor ($Arithmetic_{acc}$: $p < .001$; $BF_{inclusion} > 100$; TTA: $p = .001$; $BF_{inclusion} > 100$), not $MM_{spell}$ ($Arithmetic_{acc}$: $p = .41$; $BF_{inclusion} = 0.18$; TTA: $p = .10$; $BF_{inclusion} = 1.36$). On the other hand, when $MM_{arith}$ and $MM_{spell}$ were used to predict spelling performance, only $MM_{spell}$ was a significant and supported predictor ($Spelling_{acc}$: $p < .001$; $BF_{inclusion} > 100$; Dictation: $p < .001$; $BF_{inclusion} > 100$), not $MM_{arith}$ ($Spelling_{acc}$: $p = .38$; $BF_{inclusion} = .06$; Dictation: $p = .61$; $BF_{inclusion} = .24$).

## Interim discussion

The results of Study 1 reveal that within-domain metacognitive monitoring was an important predictor of both arithmetic and spelling performance. Monitoring measures in both domains

**Table 2. Regression analyses of $MM_{arith}$ and $MM_{spell}$ performance with metacognitive monitoring in the other domain and standardized task performance in both domains as predictors.**

|  | $MM_{arith}$ | | | |
|---|---|---|---|---|
|  | $\beta$ | $t$ | $p$ | $BF_{inclusion}$ |
| Intellectual ability | .16 | 2.12 | .04 | 2.90 |
| TTA | .26 | 3.62 | <.001 | 72.57 |
| Dictation | -.14 | -1.50 | .14 | 1.46 |
| $MM_{spell}$ | .51 | 5.26 | <.001 | >100 |
|  | $MM_{spell}$ | | | |
|  | $\beta$ | $t$ | $p$ | $BF_{inclusion}$ |
| Intellectual ability | .07 | 1.07 | .29 | 0.38 |
| Dictation | .55 | 8.77 | <.001 | >100 |
| TTA | .01 | 0.13 | .90 | 0.25 |
| $MM_{arith}$ | .34 | 5.26 | <.001 | >100 |

**Table 3. Regression analyses of arithmetic performance (i.e., arithmetic$_{acc}$ and TTA) and spelling performance (i.e., spelling$_{acc}$ and dictation) with metacognitive monitoring in the other domain and standardized task performance in the other domain as predictors.**

| | Arithmetic | | | | | | | |
|---|---|---|---|---|---|---|---|---|
| | Arithmetic$_{acc}$ | | | | TTA | | | |
| | $\beta$ | $t$ | $p$ | BF$_{inclusion}$ | $\beta$ | $t$ | $p$ | BF$_{inclusion}$ |
| MM$_{spell}$ | .54 | 5.18 | <.001 | 5.03 | .24 | 2.11 | .04 | >100 |
| Dictation | -.06 | -.54 | .59 | 2.07 | .19 | 1.73 | .09 | 0.37 |
| | Spelling | | | | | | | |
| | Spelling$_{acc}$ | | | | Dictation | | | |
| | $\beta$ | $t$ | $p$ | BF$_{inclusion}$ | $\beta$ | $t$ | $p$ | BF$_{inclusion}$ |
| MM$_{arith}$ | .47 | 5.89 | <.001 | >100 | .23 | 2.66 | .009 | 10.84 |
| TTA | .12 | 1.46 | .15 | 0.86 | .25 | 2.95 | .004 | 23.59 |

were highly correlated and predictive of one another, even after controlling for intellectual ability and performance on both academic tasks. Both monitoring measures correlated with performance in the other academic domain, ever after controlling for performance within the domain (e.g., significant correlation of MM$_{arith}$ with spelling performance, controlled for arithmetic performance). When monitoring within the domain was added above monitoring across-domain, only monitoring within the domain remained a significant predictor of academic performance. Taken together, these results provide substantial evidence for domain-generality of metacognitive monitoring in academic domains in 8-9-year-olds, in addition to the importance of some degree of domain-specificity in monitoring skills.

These results leave the question of whether this domain-generality is the result of a shift (e.g., [19]) in early primary school unanswered. One possibility is that the 8-9-year-olds already went through an important transition regarding domain-generality of metacognitive monitoring, but that such domain-generality is not observed at younger ages. On the other hand, it is possible that no shift to domain-generality has occurred because also at a younger age, domain-generality can be observed. To test this, we additionally recruited a new sample of children that were one year younger, i.e. 7-8-year-olds (Study 2). The same research questions as in Study 1 were studied using the exact same paradigm. This allowed us to test whether domain-generality is already observed at younger ages or not.

## Study 2: Metacognitive monitoring in arithmetic and spelling in 7-8-year-olds (second grade)

### Methods

**Participants.**   Participants were 77 typically developing Flemish 7–8 year-olds (second grade; 49 girls; M$_{age}$ = 7 years, 8 months; SD = 4 months; [7 years 1 month—8 years 8 months]), without a diagnosis of a developmental disorder, and who came from a dominantly middle-to-high socio-economic background. For every participant, written informed parental consent was obtained.

**Procedure.**   The procedure was the same as in Study 1.

**Materials.**   Materials were the same as in Study 1. The items in the custom arithmetic and spelling tasks were adapted from Study 1 to be age appropriate for second graders. Namely, for arithmetic, only single-digit addition was administered ($n$ = 30); for spelling only two specific Dutch spelling rules were used (i.e., extension rule and to be retrieved words with diphthongs; $n$ = 30). The standardized arithmetic task was exactly the same as in Study 1. As for the

standardized dictation, we administered the subtest for children in the middle of second grade, which includes age-appropriate, curriculum-based items [38] ($n = 42$).

**Data analysis.**   For this follow-up study, we carried out the same analyses as preregistered for Study 1 (https://osf.io/ypue4/?view_only=ce9f97af0e3149c28942a43499eafd32). The same exclusion criteria for data as in Study 1 were applied. Less than 4% of the data per task was excluded as an outlier; less than 0.90% of the items per task were excluded as a measurement error.

## Results

The descriptive statistics of all measures are presented in S1 Appendix. Additionally, Pearson correlation coefficients of all variables under study were calculated (S2 Appendix). Although not originally pre-registered, we additionally re-calculated all analyses below with chronological age as an additional control variable. Considering chronological age within grade in the analyses reported below did not change the interpretation of the results (S3 Appendix).

**The role of metacognitive monitoring in arithmetic and spelling performance.**   Pearson correlation coefficients of the associations between metacognitive monitoring and the academic skills are presented in Table 4.

$MM_{arith}$ was significantly correlated with all three arithmetic performance measures. Bayes factors indicate that the evidence for an association with $Arithmetic_{acc}$ and the TTA is decisive, while there is only anecdotal evidence for an association with $Arithmetic_{rt}$.

$MM_{spell}$ was significantly correlated with both $Spelling_{acc}$ and dictation, with Bayes factors indicating moderate to decisive evidence for an association. There was no significant correlation with $Spelling_{rt}$ and the Bayes factor indicated moderate evidence in favour of no association.

Based on the same rationale as Study 1, the response time performance measures were not considered in further analyses.

**Domain-specificity of the role of metacognitive monitoring.**   $MM_{arith}$ and $MM_{spell}$ were not significantly correlated after controlling for intellectual ability ($r = .14$, $p = .28$). The Bayes factor indicated there was moderate evidence in favour for no association ($BF_{10} = 0.28$). Hence, further control analyses (i.e., in line with Study 1 in which the correlation between $MM_{arith}$ and $MM_{spell}$ was also controlled for performance on the TTA and Dictation) were not performed.

Table 4 shows cross-domain correlations between academic performance and metacognitive monitoring in the other domain. $MM_{arith}$ was not significantly correlated with any of the spelling performance measures. Bayes factors indicated moderate evidence in favour of no association. $MM_{spell}$ was not significantly related to any of the arithmetic measures. Bayes factors indicated anecdotal to moderate evidence in favour of no association.

## Interim discussion

The results of Study 2 revealed that within-domain metacognitive monitoring was an important predictor of both arithmetic and spelling performance. Monitoring measures in both domains were not correlated, and both monitoring measures did not correlate with performance in the other academic domain. These results provide substantial evidence for domain-specificity of metacognitive monitoring in academic domains in 7-8-year-olds (second graders). No domain-general effect of metacognitive monitoring was observed, in contrast to the 8-9-year-olds (third grade children; Study 1).

**Table 4. Correlation analyses of metacognitive monitoring and academic performance measures in 7-8-year-olds (Grade 2).**

| | Arithmetic | | | Spelling | | |
|---|---|---|---|---|---|---|
| | Custom task–Accuracy[a] | Custom task—RT[b] | Standardized task (TTA)[a] | Custom task—Accuracy[a] | Custom task -RT[b] | Standardized task (dictation)[a] |
| Metacognitive Monitoring | | | | | | |
| *Arithmetic* | | | | | | |
| r | .74 | .30 | .47 | .11 | .06 | .20 |
| p | <.001 | .02 | <.001 | .38 | .66 | .11 |
| $BF_{10}$ | >100 | 2.60 | >100 | 0.23 | 0.17 | 0.53 |
| *Spelling* | | | | | | |
| r | .03 | .11 | .05 | .89 | .03 | .32 |
| p | .84 | .40 | .69 | <.001 | .82 | .01 |
| $BF_{10}$ | 0.16 | 0.11 | 0.17 | >100 | 0.16 | 4.12 |

[a] Controlled for intellectual ability.

[b] Controlled for intellectual ability and motor speed on the keyboard.

## General discussion

Two recent studies found evidence for within-domain metacognitive monitoring as an important predictor of arithmetic [5,6]. One outstanding question is whether these results regarding the role of metacognitive monitoring in arithmetic are specific to the arithmetic domain, or whether they are reflective of a more general role of metacognitive monitoring in academic performance. This study adds to the existing literature in an important way by (a) investigating metacognitive monitoring in two related, yet distinct academic domains, (b) studying whether monitoring in one domain was associated with and predictive of monitoring in the other domain (and vice versa), and (c) studying whether monitoring in one domain was associated with and predictive of performance in the other domain (and vice versa), and importantly by (d) doing this in two important age groups, namely children aged 8–9 (Study 1) and 7–8 (Study 2), who are in an important developmental phase for both academic performance and metacognition, using the exact same paradigm in both age groups and both domains.

Our results reveal that within-domain metacognitive monitoring was an important predictor of both arithmetic and spelling performance in both 8-9-year-olds (Study 1) and 7-8-year-olds (Study 2). Although metacognitive monitoring in arithmetic and spelling were highly correlated and predictive of one another in 8-9-year-olds (Study 1), they were not in younger 7-8-year-old children (Study 2). In 8-9-year-olds, but not in 7-8-year-olds, both monitoring measures correlated with performance in the other academic domain, even after controlling for performance within the domain (e.g., significant correlation of $MM_{arith}$ with spelling performance, controlled for arithmetic performance). These results provide evidence for the emergence of domain-generality of metacognitive monitoring between second and third grade (i.e., 7-9-year-olds).

Our results nicely replicate associations between metacognitive monitoring and academic performance (e.g., [5,6,11,41,42]). Combining the data of both studies, we are able to confirm the theoretically assumed development of metacognition from highly domain- and situation-specific to more flexible and domain-general with practice and experience [43]. Our results regarding a possible underlying domain-general element of metacognitive monitoring in middle primary school children (8-9-year-olds) are in line with the existing literature in older ages and/or other domains (e.g., [19–21]). For example, Schraw and colleagues [20,22] and

Veenman and colleagues [21] found evidence for domain-generality of metacognitive monitoring in adults; Geurten et al. [19] observed a shift to domain-general metacognition between 8 and 13 across the arithmetic and memory domain. Our results also show the importance of domain-specific knowledge for metacognitive performance, as was previously found in non-academic domains (i.e., soccer) by for example Löffler and colleagues [26], in very young children by Vo and colleagues [25], and in 12-year-olds in mathematics by Lingel and colleagues [28]. Our results add to this body of research that, domain-generality of metacognitive monitoring emerges between the ages of 7-to-9, yet that domain-specific knowledge and skills remain important for metacognitive monitoring, even in highly related academic domains.

Schraw and colleagues [20] note that when performance is correlated among domains (i.e., as they were in Study 1), correlated metacognitive monitoring scores (i.e., as they were in Study 1) pose no serious threat to the assumption that monitoring is domain-specific. However, when they are correlated after removing the variation attributable to performance scores, as we did using partial correlations and regression analyses, this outcome cannot be explained on the basis of domain-specific knowledge and a domain-general argument needs to be invoked. As both monitoring performances remained significantly correlated after removing the variation attributable to performance scores, our results indicate that in 8-9-year-olds (Study 1) there might be an underlying domain-general element of metacognition within both metacognitive monitoring scores. This was not observed in 7-8-year-olds (Study 2). All in all, these results point to the emergence of domain-generality of metacognitive monitoring in between second (7–8 yo) and third (8–9 yo) grade of primary school.

Our results still provide some evidence for a domain-specific element of metacognitive monitoring in 8-9-year-olds. Although metacognitive monitoring across-domain was an important predictor of performance, the associations with monitoring within-domain were significantly larger than with monitoring across-domain. Once monitoring within a domain was considered, the predictive power of monitoring across-domain was no longer significant/supported. These results suggest the continuing importance of domain-specific knowledge and skills. This domain-specific element could explain the additional predictive power of monitoring within-domain in addition to metacognitive monitoring across-domain.

Based on the important role that metacognitive monitoring was found to have in arithmetic performance [5,6], the current study investigated the domain-specificity question of metacognition by also including spelling performance. We deliberately included a different, yet correlated skill within the academic domain to thoroughly investigate the extent to which metacognition might be domain-specific. This is different from existing research, where the domain-specificity question was investigated in very distant domains. For example, Vo and colleagues [25] investigated domain-specificity in the numerical domain versus emotion discrimination. The use of spelling next to arithmetic made it possible to use the exact same paradigm to measure metacognitive monitoring and maximize the comparability of the two tasks. The fact that the computerized tasks for arithmetic and spelling were specifically designed to be as similar as possible, minimized the possibility that the results on domain-specificity of metacognition were due to differences in paradigms. By including standardized arithmetic and spelling tasks, which are not as similar to each other and measure performance in an ecologically valid way, we minimized the possibility that the results on domain-generality of metacognition were due to similarities in paradigms. While there is substantial evidence in the current studies for the emergence of domain-general metacognitive monitoring processes, the results also indicate that, even in highly related domains, domain-specific knowledge and skills are important for metacognitive monitoring in primary school children.

Although the custom arithmetic and spelling task were designed with age-appropriate items, a slight difference in task difficulty was present, with the computerized spelling tasks

being more difficult than the arithmetic tasks. Schraw and colleagues [20] pointed out that task difficulty, as a characteristic of the test environment, might have an important influence on metacognitive monitoring. They found that, with different task difficulty levels, metacognitive monitoring in adults was mostly domain-specific, yet, once tests were matched on test characteristics, monitoring was mostly domain-general. To make sure our results were not influenced by this slight difference in task difficulty, we selected, post-hoc, a subset of items per task ($n$ = 40 for Study 1; $n$ = 20 for Study 2) that were matched on task difficulty (i.e., $t$-test comparing accuracy in arithmetic and spelling selection: Study 1: $t(138)$ = 0.12, $p$ = .91; Study 2: $t(71)$ = 0.36, $p$ = .72). These post-hoc exploratory results show that our findings on metacognitive monitoring and its specificity did not change when restricting the analyses to those items that were matched in task difficulty.

Performance measures of arithmetic and spelling were accuracy in the computerized tasks, and widely-used, standardized pen-and-paper tasks. As accuracy data were a fundamental part of our metacognitive monitoring scoring, in the interpretation of the results, the largest focus should be on the standardized measures, as metacognitive monitoring was measured independently from these measures. The computerized and the standardized tasks were both age-appropriate measures, yet the standardized tasks focused less on specific items of the curriculum (i.e., only single-digit arithmetic in the computerized arithmetic task; only specific Dutch spelling rules in the computerized spelling task), for which reason they were more wide-ranged and valid measures of children's arithmetic and spelling skills. The standardized tasks were the most ecologically valid measures, assessing arithmetic and spelling performance as they are assessed in the classroom. Including these standardized tasks in the design is an essential asset of this study compared to the existing literature (e.g., [5,6]), as we were able to generalize our results from the role of metacognitive monitoring within the task, to within the domain, independently from the task in which monitoring was measured.

Although the driving mechanisms for the gradual development from domain-specificity to domain-generality of metacognitive monitoring cannot be determined on the basis of the current study, it is important to reflect on why metacognition shifts to being more domain-general around the ages 8–9. The existing literature offers some theoretical possibilities, albeit speculatively, that should be investigated in future research.

The development from more domain-specificity of metacognitive monitoring towards more domain-generality in this age group is likely reflective of a gradual transition that occurs in the development of primary school children (e.g., [33]). In early stages of this development, children's metacognitive monitoring might still be highly dependent on the (characteristics of the) specific stimuli, while over development, through experiences of failure and success, and with practice in assessing one's performance as well as in (academic) tasks, monitoring might become more generic. These hypotheses and our results can also be interpreted within the dual-process framework of metacognition (e.g., [44–46]), which Geurten et al. [19] used to interpret their findings. According to this dual-process framework of metacognition [44–46], metacognitive judgments can, on the one hand, be experience-based, i.e., based on fast and automatic inferences made from a variety of cues that reside from immediate feedback from the task and that are then heuristically used to guide decisions. As such, these metacognitive judgments are task-dependent and probably difficult to generalize across domains. On the other hand, metacognitive judgments can be information-based, i.e., based on conscious and deliberate inferences, in which various pieces of information retrieved from memory are consulted and weighted in order to reach an advised judgment. These conscious and effortful judgments are more likely to generalize to other domains. Taken together with the current results, this dual-processing model of metacognition may suggest that 7–8 year-old (second grade) children preferentially rely on automatic inferences when making judgments, while

improvements of metacognitive abilities may enable 8–9 year-old children (third grade) to rely more often on conscious and deliberate information-based processes.

Another explanation for the gradual shift from domain-specificity to domain-generality of metacognition could be that this development might be associated with the development in other general cognitive functions, such as working memory capacity or intellectual ability. For example, Veenman and colleagues [47] found that metacognitive skills develop alongside, but not entirely as part of intellectual ability. Growth in these other general cognitive functions might enable a shift from domain-specificity to domain-generality of metacognition.

Finally, the development from domain-specificity towards domain-generality might also be driven by education, as teachers instruct children on assessing their own performance, which is at first very focussed on specific tasks. Over development, children might internalise this into a semantic network of their own abilities, which in turn might generalise to other tasks and thus become more general.

It is essential to note that none of the above-mentioned hypotheses can be empirically evaluated within the current study. The focus of the current study was on *whether* a development toward domain-generality in metacognitive monitoring occurs in primary school children, in related academic domains, and, secondly *when* this occurs. The question on *how*, i.e., what mechanisms lie behind this, and *why* this is the case at this age, are important questions for future research.

Future research should also examine the question of domain-specificity of metacognition longitudinally, investigating the potential shift from domain-specificity to domain-generality in the same group of primary school children. Such a research design will allow one to investigate the directions of the associations between metacognition and academic performance and how these associations evolve over time. Furthermore, brain-imaging research in children could be very useful to investigate the question of domain-specificity of metacognition, by, for example, testing whether metacognitive abilities for different types of tasks (partially) depend on common neural structures such as the prefrontal cortex, as has been observed in adults (e.g., [23]).

To conclude, the results of this study show that metacognitive monitoring of performance is an important predictor of academic skills in primary school children. While in young primary school children (7-8-year-olds), this process is domain-specific, in slightly older children (8-9-year-olds), this is a predominantly domain-general process, in which metacognitive monitoring of performance is an important predictor of academic skills independently of the academic task and domain it is measured in, even in highly related domains. Besides depending on domain-general metacognitive processes, metacognitive monitoring remains to be dependent of domain-specific performance and knowledge. Knowing whether metacognition is rather domain-specific or domain-general, and when domain-generality emerges, is of importance for educators, as this might impact on how they provide instructions in metacognitive monitoring, namely for each task or domain separately (i.e., domain-specific metacognition) or concurrently in different tasks and domains (expecting it to transfer to new domains; domain-general metacognition).

## Supporting information

**S1 Appendix. Descriptive statistics.**
(DOCX)

**S2 Appendix. All intercorrelations.**
(DOCX)

**S3 Appendix. Analyses with chronological age.**
(DOCX)

**S1 File.**
(DOCX)

## Acknowledgments

We would like to thank all children and schools for their participation.

## Author Contributions

**Conceptualization:** Elien Bellon, Wim Fias, Bert De Smedt.

**Data curation:** Elien Bellon.

**Formal analysis:** Elien Bellon.

**Funding acquisition:** Elien Bellon, Wim Fias, Bert De Smedt.

**Investigation:** Elien Bellon, Wim Fias, Bert De Smedt.

**Methodology:** Elien Bellon, Wim Fias, Bert De Smedt.

**Project administration:** Elien Bellon.

**Resources:** Wim Fias, Bert De Smedt.

**Software:** Elien Bellon.

**Supervision:** Wim Fias, Bert De Smedt.

**Validation:** Elien Bellon, Wim Fias, Bert De Smedt.

**Visualization:** Elien Bellon.

**Writing – original draft:** Elien Bellon.

**Writing – review & editing:** Elien Bellon, Wim Fias, Bert De Smedt.

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
