## [Decision Letter · Decision Letter 0]

22 Nov 2019

PONE-D-19-26504

Metacognition across domains: Is the association between arithmetic and metacognitive monitoring domain-specific?

PLOS ONE

Dear Mrs. Bellon,

Thank you for submitting your manuscript to PLOS ONE. I have sent your manuscript to 2 expert reviewers and have now received their feedback. As you can see from their comments (at the bottom of this email), both reviewers found merit in your manuscript. Reviewer #2 notably finds your manuscript well-written and methodologically sound, and I concur with this assessment. However, both reviewers also point to issues that would need to be addressed before the manuscript can be considered for publication. Reviewer #1 points to several concerns with the theoretical framework that would need to be addressed. Both reviewers also raise some concerns about data analysis and interpretation. Notably, I agree with reviewer #2 that it would be informative to consider age in your analyses. Considering these comments, I would invite you to revise your manuscript and submit it for further consideration by the journal. 

We would appreciate receiving your revised manuscript by Jan 06 2020 11:59PM. To enhance the reproducibility of your results, we recommend that if applicable you deposit your laboratory protocols in protocols.io, where a protocol can be assigned its own identifier (DOI) such that it can be cited independently in the future. For instructions see: http://journals.plos.org/plosone/s/submission-guidelines#loc-laboratory-protocols

We look forward to receiving your revised manuscript.

Kind regards,

Jérôme Prado

Academic Editor

PLOS ONE

Reviewers' comments:

Reviewer #1: The goal of this research was to investigate the domain specificity of metacognitive monitoring by examining the relation between metacognitive monitoring and performance across two academic domains, arithmetic and spelling, in 7- to 8- and 8- to 9-year-olds. The authors found that (a) as expected, metacognitive monitoring was correlated with performance within each domain in both age groups; (b) metacognitive monitoring in arithmetic was correlated with metacognitive monitoring in spelling only in 8-to 9-year-olds; and (c) metacognitive monitoring in one domain predicted performance in the other domain only in 8-to 9-year-olds. The authors concluded that metacognitive monitoring transitions from domain specific to domain general between 7 and 9 years of age.

Overall, the findings provided insights into the development of metacognitive monitoring in primary school children. However, there are several concerns as detailed below that should be addressed.

Major issues

Even though the study included two metacognitive monitoring measures (arithmetic and spelling), the authors often talked about them as if they are one variable, and this may cause confusions on which or both metacognitive monitoring predicted performance in arithmetic vs. spelling. For example, in the abstract, the authors stated that “Pre-registered analyses revealed that metacognitive monitoring was an important predictor of both arithmetic and spelling at both ages.” In this case, I think the authors were referring to the predictive relation between metacognitive monitoring and performance within the domains of arithmetic and spelling respectively, but that’s unclear from the sentence. An alternative interpretation is that metacognitive monitoring in general (as one variable) predicted arithmetic and spelling performance. Similar issues were present on page 7 line 11 (“(a) the associations between metacognitive monitoring and arithmetic and spelling”), on page 23 in interim discussion (“metacognitive monitoring was an important predictor of both arithmetic and spelling performance”), and on page 24 (“metacognitive monitoring was an important predictor of both arithmetic and spelling performance….”)

In the introduction, the authors introduced two aspects of metacognition: declarative knowledge and procedural metacognition, proposed by Flavell (page 3). However, these two constructs were not clearly defined until page 6. Specifically, it is unclear what declarative knowledge is, how it is different from procedural metacognition, and which construct is studied in the cited papers. Moving the definitions and examples of these two constructs from page 6 to page 3, and be clearer about describing the metacognitive measures used in the cited studies would help resolve the confusions.

Components of metacognition were introduced in paragraph 2 of the introduction. However, it is difficult to imagine and differentiate these components of metacognition in the current writing. For instance, it is unclear what “regulating knowledge (introduction, paragraph 2, line 5)” would look like. What is an example of regulating knowledge? It is also unclear how “knowing the limits (line 4)” is different from “monitoring knowledge (line 5)”. In this paragraph, the authors suggested that two components of metacognition are important for self-regulated and successful learning (lines 5-6). However, there seems to be three different components: knowing the limits, monitoring knowledge, and regulating knowledge. Providing examples and/or descriptions would help clarify these confusions.

On page 6, the authors suggested that there are distinctions between metacognitive monitoring and metacognitive control. However, it is unclear what metacognitive monitoring and metacognitive controls are, and how they are different. Furthermore, the authors referenced Bellon et al to support this distinction between monitoring vs. control, but Bellon et al focused on metacognitive monitoring and declarative metacognitive knowledge. Is declarative metacognitive knowledge an aspect of metacognitive control?

On page 7, the authors listed four points for research questions. However, it seemed that there were only two explicit predictions (i and ii) mapping onto research question (b) and (c). Research questions (a) and (d) did not have clear predictions associated with them. Aligning the research questions with their corresponding predictions would help make the paper easy to follow.

Authors predicted that metacognitive monitoring would be domain general in 8-9-year-olds and domain specific in 7-8-year-olds (page 8). Based on the introduction, it seemed that there is evidence supporting the first prediction (Geurten et al), but it is less clear why the authors believed metacognitive monitoring would be domain specific in the younger age group. Even though Vo et al provided preliminary support for the second prediction in 5- to 8-year-olds, the study was on two very different domains (numerical vs. emotion), thus it is unclear why the authors would think the same pattern of domain specificity would emerge in comparable academic domains (arithmetic vs. spelling).

The authors mentioned that there were four practice trials for each of the computer tasks (page 10 line 8), but it is unclear whether there were four practice trials for addition problems and four practice trials for multiplication problems, or there was a total of four practice trials in the arithmetic computer task. Were the practice trials in block 1, block 2, or both? Were the practice trials included in the accuracy and response time measures of task performance? Similar questions need to be addressed for the spelling task.

The BF cutoffs for the alternative hypothesis (page 14) were extremely helpful in guiding interpretations of the findings. Perhaps it would be helpful to provide BF cutoffs for null hypothesis as well. Based on the BF descriptions, BF seems to be the ratio of the evidence for alternative vs. null hypotheses, so is it correct to infer that a BF value between 1/10 and 1/3 suggests moderate support for null hypothesis (because 3<bf<10 alternative="" for="" hypothesis="" moderate="" provides="" support="" the="">

The authors stated that they investigated the unique role of metacognitive monitoring in academic performance in the results section (page 17). Although this seemed to be related to the research question 3: whether performance in one domain is associated with and/or predicted by metacognitive monitoring in the other domain, the statement in the result section did not suggest cross domain prediction as stated in the introduction. Furthermore, the two models including both MMarith and MMspell as predictors of arithmetic and spelling performance respectively (page 19) were not mentioned prior to the reporting of the results. It may be helpful to outline the specific models in the methods or the beginning of the results section to help guide readers’ expectation.

The authors utilized regressions to control for Intelligence, Dictation, and TTA when examining the association between MMarith and MMspell. I wonder why the accuracy on the two computer tasks were not controlled for in these regression models. Is it because of the high correlations between MM and accuracy on the computer tasks (rs>.89), and the concern of collinearity in predictors? I think the high correlations between MM and accuracy raises another question on whether metacognitive monitoring is really different from performance. While I understand that the two constructs were measured differently, I wonder how much between-item variations there were on the MM questions within an individual. Because it seems that participants could take as long as they needed to choose their answer on the computer tasks, they could be highly accurate on the computer tasks (as suggested by accuracy especially for the arithmetic task), and responded “correct” on all MM questions. In this case, there were little to no variation in the MM questions so did the participants really calibrate their confidence based on their performance/knowledge or did they just blindly select “correct” on all MM questions?

The authors controlled for IQ, and the scores on TTA and Dictation when examining the correlation between MMarith and MMspell in 8-9-year olds, but only controlled for IQ when examining the correlation between MMarith and MMspell in 7-8-year olds. While I understand that the correlation between MMarith and MMspell in 7-8-year olds was already not significant when only controlling for IQ, it may still be helpful to be more explicit and consistent on the different analytic approaches for the two age groups.

Minor issues

“On one hand” should precede the phrase “On the other hand”. If the first hand is not present, the second hand is not really “the other hand”. The authors frequently use “on the other hand” without its preceding partner, “on one hand”. (e.g., page 4 line 8, page 5 line 16, page 27 line 1, page 28 line 15) Please review the paper and adjust the phrasing.

The sentence “Materials consisted of standardized tests, paper-and-pencil tasks, and computer tasks…. (Page 9, Materials)” seemed to suggest that there were three types of tasks, and standardized tests were different from paper and pencil tasks. However, the standardized tests seemed to be the paper-and-pencil tasks. I would suggest rephrasing the sentence to “Materials consisted of standardized written tests and custom computer tasks….”

Although the authors stated that the computer tasks were arithmetic or spelling verification (page 9 line 8), the task descriptions suggested that the children were choosing the right answer (8+2 is 10 or 16) rather than verifying the answer (8+2=16, is the answer correct?). It would help avoid confusions by not characterizing them as verification tasks.

In tables 2 and 3, some ts and ps are in uppercase. I think they should all be lowercase.

Reviewer #2: Review of “Metacognition across domains: Is the association between arithmetic and metacognitive monitoring domain-specific?”

The authors present the results from two nearly identical studies, one with third graders (ages 8-9) and one with second graders (ages 7-8). The children completed a computerized arithmetic task, a standardized arithmetic task, a computerized spelling task, and a standardized spelling task. During the computerized tasks, children also rated their certainty on a large portion of the items. Finally, children completed an IQ task and a motor fluency task as control variables.

In the third-grade sample, the authors provide evidence for domain-general metacognition. Monitoring scores in arithmetic were associated with arithmetic skills, spelling skills, and spelling monitoring scores. Also, monitoring scores in spelling were associated with spelling skills, arithmetic skills, and arithmetic monitoring scores. In contrast, in the second-grade sample, the authors provide evidence for domain-specific metacognition. Monitoring scores in arithmetic were only associated with arithmetic skills, but not with spelling skills or spelling monitoring scores. Also, monitoring scores in spelling were only associated with spelling skills, but not with arithmetic skills or arithmetic monitoring scores.

The reported paper has many strengths, including the use of two samples of different ages, monitoring data in multiple domains (arithmetic and spelling), and two different skills tests in each domain (one on which monitoring was also measured and one that was standardized). The analyses answer a critical question in the literature and make a novel contribution. The design is technically sound, the writing is clear, and the claims appear supported by the data.

I have several comments for the authors to consider.

First, the conclusions will be better supported if the authors provide additional potential explanations for the different results across ages. The current discussion highlights how their results are consistent with previous work. For example, they write, “we are able to confirm the theoretically assumed development of metacognition from highly domain- and situation specific to more flexible and domain-general with practice and experience.” However, the current results suggest this may be tied to a fairly narrow time frame (between the ages of 7 and 9). Why does metacognition shift to being more domain-general? Why does this occur around ages 8 and 9? What kinds of practice and experience are theorized to be related to this shift? Additional insights into why this shift occurs around this age would help situate the novel empirical findings into the broader theoretical landscape of metacognition.

Second, I was a bit surprised that age was not featured in the analyses at all. For example, within each study, children’s ages spanned a full year (e.g., ranging from 8 years, 4 months to 9 years, 4 months in Study 1). It seems reasonable to investigate whether age is correlated with the other metrics (e.g., arithmetic skills, metacognitive monitoring) and potentially control for any shared variance across them related to age. Also, an interesting aspect of the studies is that there is an overlap in children’s ages across the studies, despite the children being in different grades. Specifically, it appears that some children in Study 1 and some children in Study 2 are between 8 years, 4 months and 8 years, 8 months. The authors may be able to provide additional insight into this metacognitive “shift” by potentially performing supplemental analyses on 8s vs. 9s in Study 1 and 7s vs. 8s in Study 2. I realize the authors pre-registered their analyses, which is 100% desirable and laudable, but also means any analyses with age would be considered exploratory or supplemental. I would encourage the authors to consider additional analyses with age in the models. At the very least, the authors should provide a justification in the paper for the reasons they opted not to include age in their tables and models.

Third, I was also a bit surprised by the lack of attention to characterizing children’s metacognition more broadly at these ages. The authors provide basic descriptive statistics (means, standard deviation, and range) in the supplemental materials. In general, children’s metacognitive monitoring seems to be quite good, with average calibration scores around 1.4 to 1.8 (out of 2). However, additional information could help shed light on how children are performing on this task. For example, regardless of their performance, how often do children think that they are correct vs. how often do children select that they do not know? Similarly, are average calibration scores higher on correct responses or incorrect responses? When children are “uncalibrated,” is it more often because they are overconfident (thinking they are correct when actually wrong) or because they are underestimating their skills (thinking they are incorrect when actually right). Do these metrics vary by discipline? These findings would not change current conclusions about domain-specificity, but would provide additional contributions by better characterizing children’s metacognitive monitoring on these tasks.

Fourth, on a very minor note, I found two pieces of the method section to be a bit confusing. First, when describing the procedure, the authors write, “The participants completed all tasks in the same order in an individual session, two sessions in small groups of eight children and a group session in the classroom.” I assume this means each child participated in four sessions. Is this because some tasks needed to be assessed one-on-one and other tasks did not? I think it would help to clarify which tasks were administered in which sessions and to clarify the timing of these sessions (e.g., after the individual session, how many days later was the small group session? What time of the school year were these sessions? Etc.). Second, the authors describe the computerized arithmetic and computerized spelling tasks as “verification” tasks. This made me assume that a single problem/word was presented and the child had to verify (click yes or no) as to whether it was correct. However, in reality, the task included two simultaneous presentations of a problem/word, one that was correct and one that was incorrect. The child had to select the correct one. This is super minor, but it might be more appropriate to call it a selection task or recognition task rather than a verification task for ease of interpretation.

 </bf<10>

---

## [Author Response · Author response to Decision Letter 0]

7 Jan 2020

PONE-D-19-26504

Metacognition across domains: Is the association between arithmetic and metacognitive monitoring domain-specific?

Dear Editor and Reviewers,

We thank you for your careful reading and thoughtful comments on our manuscript. We appreciate the time and effort that you have dedicated to providing this valuable feedback. We have taken your comments into account in our revision, which resulted in a manuscript that is in our opinion clearer and more compelling. Please find below our point-by-point response to your comments and queries. To ease the identification of changes to the text in the revised manuscript, we have highlighted all changes by using coloured text. We look forward to hearing from you in due time regarding our submission and to respond to any further questions and comments you may have. We hope you will consider this manuscript for publication in PLOS ONE.

Responses to the Editor

Editor, general point

Thank you for submitting your manuscript to PLOS ONE. I have sent your manuscript to 2 expert Reviewers and have now received their feedback. As you can see from their comments (at the bottom of this email), both Reviewers found merit in your manuscript. Reviewer #2 notably finds your manuscript well-written and methodologically sound, and I concur with this assessment. However, both Reviewers also point to issues that would need to be addressed before the manuscript can be considered for publication. Reviewer #1 points to several concerns with the theoretical framework that would need to be addressed. Both Reviewers also raise some concerns about data analysis and interpretation. Notably, I agree with Reviewer #2 that it would be informative to consider age in your analyses. Considering these comments, I would invite you to revise your manuscript and submit it for further consideration by the journal.

Author’s response: We thank the Editor and the Reviewers for their positive evaluation of the manuscript and their constructive feedback on our theoretical framing and data-analysis. We agree that these were areas with room for improvement and we have revised the manuscript in accordance with these comments. Specifically, we addressed the concerns of Reviewer 1 regarding the theoretical framework (see Reviewer 1 points 1-5). We also considered age as a variable in the analyses (Reviewer 2, point 2) revealing that age was not correlated with any of the performance measures and that including chronological age in the analyses did not change the interpretation of the current results. In the remainder of this response letter, we provide a point-by-point response to all issues raised by the two Reviewers.

Responses to Reviewer 1

Reviewer 1, point 1

Even though the study included two metacognitive monitoring measures (arithmetic and spelling), the authors often talked about them as if they are one variable, and this may cause confusions on which or both metacognitive monitoring predicted performance in arithmetic vs. spelling. For example, in the abstract, the authors stated that “Pre-registered analyses revealed that metacognitive monitoring was an important predictor of both arithmetic and spelling at both ages.” In this case, I think the authors were referring to the predictive relation between metacognitive monitoring and performance within the domains of arithmetic and spelling respectively, but that’s unclear from the sentence. An alternative interpretation is that metacognitive monitoring in general (as one variable) predicted arithmetic and spelling performance. Similar issues were present on page 7 line 11 (“(a) the associations between metacognitive monitoring and arithmetic and spelling”), on page 23 in interim discussion (“metacognitive monitoring was an important predictor of both arithmetic and spelling performance”), and on page 24 (“metacognitive monitoring was an important predictor of both arithmetic and spelling performance….”).

Author’s response: We thank the Reviewer for highlighting this issue and apologize for being unclear on this matter. We have added the specification “within-domain metacognitive monitoring” in all these statements referred to by the Reviewer (i.e., abstract, pages 7, 23 and 24) to clarify that the statements indeed indicate the predictive relation between metacognitive monitoring and performance within the domain in which metacognitive monitoring was measured. In addition to having revised the wording of the sentences pointed to by the Reviewer in the abstract, on pages 7, 23 and 24, we have checked the entire manuscript for sentences which might have the same issue and adjusted these accordingly, i.e. on pages 6, 18, 20 and 25.

Reviewer 1, point 2

In the introduction, the authors introduced two aspects of metacognition: declarative knowledge and procedural metacognition, proposed by Flavell (page 3). However, these two constructs were not clearly defined until page 6. Specifically, it is unclear what declarative knowledge is, how it is different from procedural metacognition, and which construct is studied in the cited papers. Moving the definitions and examples of these two constructs from page 6 to page 3, and be clearer about describing the metacognitive measures used in the cited studies would help resolve the confusions.

Author’s response: We fully agree with the Reviewer that it makes more sense to define the different aspects of metacognition in the beginning of the introduction, when they are first mentioned. Therefore, following the Reviewers suggestion, we moved the definitions originally stated on page 6, to page 3 to make this more clear for the reader. Following another point raised by the Reviewer (see Reviewer 1, point 4), we additionally focussed on the critical component that we investigated in our manuscript, i.e., metacognitive monitoring, by defining it in more detail in the first paragraph of the manuscript (i.e., “An important aspect of procedural metacognition is metacognitive monitoring, which is defined as the subjective self-assessment of how well a cognitive task will be/is/has been performed [1,2]”; see page 3). We agree with the Reviewer that this resulted in a more clear-cut introduction. On the other hand, we would like to point out that the distinction between declarative and procedural metacognition is and cannot always be carefully made in the different studies we are citing in the introduction. In an attempt to make the text as comprehensible as possible, we have clarified and included these distinctions wherever possible.

Reviewer 1, point 3

Components of metacognition were introduced in paragraph 2 of the introduction. However, it is difficult to imagine and differentiate these components of metacognition in the current writing. For instance, it is unclear what “regulating knowledge (introduction, paragraph 2, line 5)” would look like. What is an example of regulating knowledge? It is also unclear how “knowing the limits (line 4)” is different from “monitoring knowledge (line 5)”. In this paragraph, the authors suggested that two components of metacognition are important for self-regulated and successful learning (lines 5-6). However, there seems to be three different components: knowing the limits, monitoring knowledge, and regulating knowledge. Providing examples and/or descriptions would help clarify these confusions.

Author’s response: We entirely understand the concerns raised by the Reviewer and apologize for this ambiguous sentence. The aim of this sentence was not to echo the different aspects of metacognition defined earlier in the introduction, but to give examples of possible underlying mechanisms through which metacognition may play an important role in different (cognitive) domains. For example, the regulation of cognition refers to metacognitive activities that help to control one’s thinking or learning. Different regulatory skills have been described in the literature, but it typically refers to three components: planning, monitoring, and evaluation [3]. An example of regulating knowledge referred to in the next sentence in the manuscript is “checking answers when one feels unsure about the correctness of the answer”. We now see that the addition of “monitoring knowledge” on top of “regulating knowledge” in the original manuscript is confusing, as monitoring is an integral part of regulating knowledge. We fully agree that by adding the quantifier “two” (i.e., “two essential components”) we made the interpretation of this sentence unnecessarily complicated. To make this sentence more clear, we have rephrased it (see below), making the connection more transparent between the examples of metacognitive aspects (i.e., knowing the limits of your knowledge and regulating that knowledge), self-regulated and successful learning in different domains and the examples given in the next sentence (e.g., allocating study time). 

“The importance of metacognition that was found in existing research in different (cognitive) domains is not surprising, as metacognitive aspects such as knowing the limits of your own knowledge and being able to regulate that knowledge, are essential components of self-regulated and successful learning [4], enabling learners to improve their cognitive achievements. For example, good metacognition allows learners to correctly allocate study-time, check answers when they feel unsure about the correctness of the answer or provide a learning moment when an error is detected.” (manuscript page 3-4)

Additionally, and in line with above and below mentioned comments by the Reviewer (see Reviewer 1, point 2 and Reviewer 1, point 4), we have now included the definitions of the different components of metacognition at the beginning of the manuscript, and added a more detailed definition of metacognitive monitoring in the first paragraph of the manuscript, such that these concepts are already introduced to the reader before the abovementioned paragraph 2. 

Reviewer 1, point 4

On page 6, the authors suggested that there are distinctions between metacognitive monitoring and metacognitive control. However, it is unclear what metacognitive monitoring and metacognitive controls are, and how they are different. Furthermore, the authors referenced Bellon et al to support this distinction between monitoring vs. control, but Bellon et al focused on metacognitive monitoring and declarative metacognitive knowledge. Is declarative metacognitive knowledge an aspect of metacognitive control?

Author’s response: Following the second point of the Reviewer (see Reviewer 1, point 2), we have now revised the paragraph on the definition of metacognition and its different aspects (i.e. paragraph 1, page 3 – see below). This rearrangement additionally tackled the confusion concerning the reference to Bellon et al, which indeed does not investigate metacognitive control (i.e., an aspect of procedural metacognition), but focusses on declarative metacognition and metacognitive monitoring. 

We completely agree that it causes confusion to name metacognitive control without further explaining it, and, importantly, without this being of interest in the remainder of the manuscript. Therefore, we have removed the sentence on the distinction between metacognitive monitoring and control (i.e., which was originally in the first paragraph of the introduction), and now focussed on the critical component we investigated in our manuscript, i.e., metacognitive monitoring. Additionally, we have defined metacognitive monitoring in more detail in the first paragraph of the manuscript. This resulted, in our opinion, in a more clear-cut introduction in which the focus is on the critical components of the current study. These are now also explained in more detail compared to the original manuscript.

Manuscript page 3, paragraph 1: 

“… Noticing your mistakes, an example of monitoring your cognition, is a facet of metacognition, a concept first introduced by Flavell [5] as a broader concept that encompasses on the one hand declarative, metacognitive knowledge (i.e., the ability to assess one’s own cognitive knowledge and ability, knowledge about cognition and learning) and on the other hand, procedural metacognition (i.e., self-reflecting, higher-order cognitive processes, in other words, how people monitor and control their cognition during ongoing cognitive processes) [1,6]. An important aspect of procedural metacognition is metacognitive monitoring, which is defined as the subjective self-assessment of how well a cognitive task will be/is/has been performed [1,2].”

Reviewer 1, point 5

On page 7, the authors listed four points for research questions. However, it seemed that there were only two explicit predictions (i and ii) mapping onto research question (b) and (c). Research questions (a) and (d) did not have clear predictions associated with them. Aligning the research questions with their corresponding predictions would help make the paper easy to follow.

Author’s response: We thank the Reviewer of pointing to this important issue. The current manuscript indeed investigated whether metacognitive monitoring is domain-specific, by means of four research questions, namely:

“investigating (a) the associations between within-domain metacognitive monitoring and arithmetic and spelling; (b) whether metacognitive monitoring in one domain is associated with and/or predicted by metacognitive monitoring in the other domain; (c) whether performance in one domain is associated with and/or predicted by metacognitive monitoring in the other domain, and (d) these questions in two different age groups in primary school to fully grasp potential transitional periods in the domain-specificity of metacognitive monitoring” (manuscript page 7).

However, we have wrongly stated in the paragraph following these research questions (i.e., manuscript page 7) that predictions would be made. The statements under (i) and (ii) are not predictions in itself, but are meant to provide theoretical background or interpretation guidelines for the reader, as in these statements we describe which outcome would point to which interpretation (i.e., domain-specificity vs. domain-generality of metacognitive monitoring). We apologize for being unclear in the original manuscript. We have rephrased this paragraph (see below; manuscript page 7), removing the reference to prediction.

“If, on the one hand, metacognition is highly domain-general, then metacognitive monitoring in the arithmetic and spelling tasks will be correlated and predictive of each other, even when controlled for academic performance – as arithmetic and spelling are highly related domains; and metacognitive monitoring in one domain will be associated with and predicts academic performance in the other domain. If, on the other hand, metacognition is highly domain-specific, then the associations described above will be non-significant (frequentist statistics) and Bayes factors will be close to zero (Bayesian statistics; see below).”

Reviewer 1, point 6

Authors predicted that metacognitive monitoring would be domain general in 8-9-year-olds and domain specific in 7-8-year-olds (page 8). Based on the introduction, it seemed that there is evidence supporting the first prediction (Geurten et al), but it is less clear why the authors believed metacognitive monitoring would be domain specific in the younger age group. Even though Vo et al provided preliminary support for the second prediction in 5- to 8-year-olds, the study was on two very different domains (numerical vs. emotion), thus it is unclear why the authors would think the same pattern of domain specificity would emerge in comparable academic domains (arithmetic vs. spelling).

Author’s response: Both predictions formulated in this paragraph are explicitly based on Geurten, Meulemans, and Lemaire (2018), who predicted that starting around the age of 8 there would be a shift from domain-specific to domain-general metacognition, and who observed in their data that while metacognition is first domain-specific, a gradual development from domain-specific to domain-general metacognition occurs in children when they are between 8 and 13 years old. To investigate this proposed development from domain-specificity before the age of 8 to the gradual transition to domain-generality between the age of 8 and 13 years, we specifically selected two age groups. Namely, one group was specifically chosen just under this age-range, i.e., 7-8-year-old group. In this age-group, based on the suggestion of Geurten et al., it is predicted that children’s metacognitive monitoring is domain-specific. The other age group, i.e. 8-9-year-olds, is exactly at the beginning of the age range for which, based on the predictions and results of Geurten et al. (2018), domain-generality is starting to emerge. 

We specifically recruited two age-groups that only differed in one grade, such that the same arithmetic, spelling and metacognitive monitoring paradigm could be used, in order to maximize comparability between age-groups. Moreover, two highly related academic domains were chosen, to ensure a stringent empirical test of the possible limits of domain-specificity of metacognitive monitoring. Indeed, as the Reviewer suggests, it is plausible that domain-specificity in children is much harder to ascertain in related domains, in contrast to Vo, Li, Kornell, Pouget, and Cantlon (2014), who demonstrated this in very different domains, for which such specificity might on a surface level be more easily observed. However, a very stringent empirical investigation of the domain-specificity of metacognitive monitoring was exactly the aim of the current study, for which reason we specifically selected two very closely related domains. As a result, we contend that our data provide a stronger test of domain-specificity compared to previously reported work. 

Reviewer 1, point 7

The authors mentioned that there were four practice trials for each of the computer tasks (page 10 line 8), but it is unclear whether there were four practice trials for addition problems and four practice trials for multiplication problems, or there was a total of four practice trials in the arithmetic computer task. Were the practice trials in block 1, block 2, or both? Were the practice trials included in the accuracy and response time measures of task performance? Similar questions need to be addressed for the spelling task.

Author’s response: Thank you for highlighting this ambiguity and we apologize for being unclear about this. Following the Reviewers suggestion, we have specified for both the arithmetic (manuscript page 9-10 – see below) and spelling (manuscript page 11 – see below) task that four practice items were presented before each block of the arithmetic task and six practice items before each block of the spelling task. Additionally, we clarified that performance on the practice items was not included in the performance measures (manuscript page 10 for arithmetic and 11 for spelling).

Manuscript page 10 – arithmetic task:

“Each block was preceded by four practice trials to familiarize the child with the task requirements. Performance on the practice items was not included in the performance measures.”

Manuscript page 11 – spelling task:

“Each block was preceded by six practice trials to familiarize the child with the task requirements. Performance on the practice items was not included in the performance measures.”

Reviewer 1, point 8

The BF cutoffs for the alternative hypothesis (page 14) were extremely helpful in guiding interpretations of the findings. Perhaps it would be helpful to provide BF cutoffs for null hypothesis as well. Based on the BF descriptions, BF seems to be the ratio of the evidence for alternative vs. null hypotheses, so is it correct to infer that a BF value between 1/10 and 1/3 suggests moderate support for null hypothesis (because 3<bf).

Author’s response: We thank the Reviewer for the appreciation of the described Bayes Factor interpretation guidelines. The assumption made by the Reviewer is correct: a BF10 value between 1/10 and 1/3 indeed suggests moderate support for the null hypothesis. In accordance with the suggestion by the Reviewer, we have now added the interpretation guidelines for interpreting evidence in favour of the null hypothesis in the manuscript on page 14. 

Manuscript page 14: 

“Although Bayes factors provide a continuous measure of degree of evidence, there are some conventional approximate guidelines for interpretation ([9] for a classification scheme): BF10 = 1 provides no evidence either way, BF10 > 1 anecdotal, BF10 > 3 moderate, BF10 > 10 strong, BF10 > 30 very strong and BF10 >100 decisive evidence for the alternative hypothesis; BF10 < 1 anecdotal, BF10 < 0.33 moderate, BF10 < 0.10 strong, BF10 < 0.03 very strong and BF10 < 0.01 decisive evidence for the null hypothesis.”

Reviewer 1, point 9

The authors stated that they investigated the unique role of metacognitive monitoring in academic performance in the results section (page 17). Although this seemed to be related to the research question 3: whether performance in one domain is associated with and/or predicted by metacognitive monitoring in the other domain, the statement in the result section did not suggest cross domain prediction as stated in the introduction. Furthermore, the two models including both MMarith and MMspell as predictors of arithmetic and spelling performance respectively (page 19) were not mentioned prior to the reporting of the results. It may be helpful to outline the specific models in the methods or the beginning of the results section to help guide readers’ expectation.

Author’s response: Thank you for highlighting this need for clarification. The statement on page 17 to which the Reviewer refers (i.e., “investigated the unique role of metacognitive monitoring in academic performance”), indeed pointed to the third research question discussed on page 7 in the manuscript. We now see that this wording might have been confusing for the reader as the points described under (b) and (c) on page 17 of the manuscript point to the same research question, namely “whether performance in one domain is associated with and/or predicted by metacognitive monitoring in the other domain”. To better align the research questions described in the introduction with our results section, we have rephrased this paragraph (manuscript page 17-18). Following the Reviewers suggestion, we have made the use of cross domain prediction (as stated in the introduction on page 7) more explicit in the result section (manuscript page 17-18). To avoid further confusion, we have removed the enumerations in the result section, as they did not map the enumerations of the research questions presented in the introduction. 

As the Reviewer suggested, in order to further guide the readers’ attention, we specified an outline of the different models that were used in the beginning of the results section on the domain-specificity of the role of metacognition (manuscript page 17-18).

This all results in the following paragraph (manuscript page 17-18): 

“To examine domain-specificity of the role of metacognition, we first investigated the association between MMarith and MMspell with correlation and regression analyses. Specifically, we investigated whether MMarith and MMspell were correlated, even when controlling for intellectual ability and academic performance in both domains. Controlling for intellectual ability and performance in both standardized academic tasks was necessary, to make sure the observed associations between MMarith and MMspell were not (entirely) driven by their shared reliance on intellectual ability or by the high correlation between both academic domains.

Secondly, we studied the role of MMspell in arithmetic performance and MMarith in spelling performance with correlation and regression analyses. In other words, cross-domain correlations between academic performance in one domain and metacognitive monitoring in the other domain were calculated. As performance in the arithmetic and spelling tasks was highly correlated, the cross-domain associations of metacognitive monitoring and academic performance might rely on the correlation between the academic tasks. Therefore, we used regression models to investigate whether metacognitive monitoring in arithmetic uniquely predicted spelling performance on top of arithmetic performance, and vice versa.

In a final step, we investigated the unique contribution of cross-domain metacognitive monitoring to performance over within-domain metacognitive monitoring using regression models including metacognitive monitoring in both domains as predictors for academic performance.”

Reviewer 1, point 10

The authors utilized regressions to control for Intelligence, Dictation, and TTA when examining the association between MMarith and MMspell. I wonder why the accuracy on the two computer tasks were not controlled for in these regression models. Is it because of the high correlations between MM and accuracy on the computer tasks (rs>.89), and the concern of collinearity in predictors? I think the high correlations between MM and accuracy raises another question on whether metacognitive monitoring is really different from performance. While I understand that the two constructs were measured differently, I wonder how much between-item variations there were on the MM questions within an individual. Because it seems that participants could take as long as they needed to choose their answer on the computer tasks, they could be highly accurate on the computer tasks (as suggested by accuracy especially for the arithmetic task), and responded “correct” on all MM questions. In this case, there were little to no variation in the MM questions so did the participants really calibrate their confidence based on their performance/knowledge or did they just blindly select “correct” on all MM questions?

Author’s response: In our preregistered analyses, we indeed did not include accuracy on the custom tasks when examining the association between MMartih and MMspell. While we understand the Reviewers question regarding the inclusion of the accuracy in the custom tasks, we a priori specifically selected the standardized tasks to control for academic performance because these tasks were the most (ecologically) valid and reliable measures of academic performance more broadly construed. Importantly, these standardized tests were administered independently from the metacognitive monitoring measures, which was not the case for academic performance in the custom tasks. Additionally, based on the results of the current study and as the Reviewer suggested, the high correlation between accuracy in the custom task and metacognitive monitoring make these custom tasks less suitable for simultaneous consideration in a regression model.

Although we think that, based on the different abovementioned arguments, these analyses are not eligible to be included in the manuscript, we have examined whether or not including accuracy on the custom tasks in the regression models instead of performance on the standardized academic tasks changed the interpretation of the results. As can be seen below (Table R1), this was not the case. 

 

Table R1. Regression analyses of MMarith and MMspell performance with metacognitive monitoring in the other domain and accuracy on the custom academic tasks in both domains as predictors.

 MMarith

 β t p BFinclusion

Intellectual ability .07 1.43 .15 .67

Custom task Arithmetic – Accuracy .76 15.41 <.001 >100

Custom task Spelling – Accuracy -.27 -2.44 .02 2.32

MMspell .41 3.63 <.001 40.68

 MMspell

 β t p BFinclusion

Intellectual ability -.003 -.08 .91 0.12

Custom task Spelling – Accuracy -.10 -1.65 .10 0.34

Custom task Arithmetic – Accuracy .87 23.24 <.001 >100

MMarith .22 3.63 <.001 41.63

Because the results of the analyses including academic achievement as measured with the standardized academic tasks and the results including academic achievement as measured with accuracy in the computerized academic tasks were the same, and because controlling for the standardized tasks is more suitable from a theoretical point of view (see arguments above), we decided to adhere to our original pre-registered analyses which include the standardized academic tasks. However, to be fully transparent in the manuscript, we added the following statement on page 18: “Additional post-hoc analyses that were not preregistered indicated that the results were the same when including academic achievement as measured with accuracy in the computerized academic tasks instead of academic achievement as measured with the standardized academic tasks.”

Turning to the Reviewers concern regarding the difference between metacognitive monitoring and performance, which we understand, there are several arguments and observations that confirm that there is most certainly a difference between monitoring and performance. A first indication that these are indeed two different variables can be found in the abovementioned regression analyses, in which it is shown that metacognitive monitoring is predictive of metacognitive monitoring in the other domain, even in addition to accuracy in that task. This indicates that the metacognitive monitoring measure explains unique variance that is not accounted for by accuracy in either arithmetic or spelling. 

Secondly, within the existing literature (e.g., [10,11], it has been consistently shown that performance on a cognitive task can be distinguished from metacognitive monitoring in that task. This was specifically found for retrospective confidence judgements (e.g., [10,12]), which is how metacognitive monitoring was operationalised in the current study. For example, fMRI studies comparing activity during task performance versus retrospective metacognitive judgements found important brain activation differences between task performance and metacognitive monitoring (e.g., Chua, Schacter, Rand-Giovannetti, & Sperling, 2006; Chua, Schacter, & Sperling, 2009). Lesion studies also confirm the difference between task performance and metacognitive performance by showing, for example, that patients with parietal lesions may have impairments in retrospective metacognitive performance despite little or no impairment in accompanied task performance (e.g., Berryhill, 2012; Davidson et al., 2008; Simons, Peers, Mazuz, Berryhill, & Olson, 2010). 

As correctly inferred by the Reviewer, there was indeed no time limit on providing an answer in the arithmetic and spelling tasks. However, it is critical to emphasize that children were specifically instructed to answer as quickly as possible. Moreover, the participating children were all used to providing academic answers as fast and accurate as possible, due to a curricular focus on fluency. Additionally, as preregistered, items in which response time for academic answers was more than three standard deviations from the mean on both subject level and item level, were excluded from the data-analysis. Data on general task performance of children for who this task performance was more than three standard deviations from the mean of the task, were also excluded from the data-analysis, as preregistered. These preregistered exclusion criteria further ensure that items on which children took excessively long to provide an academic answer and task performance of children who in general took excessively long to provide an academic answer, were discarded. 

It is indeed important to note, as the Reviewer pointed to, that while children were highly accurate on the academic tasks, there still was between-item variation in the answers to the metacognitive monitoring question. As also outlined in our Response to Reviewer 2, point 3, regardless of performance, in both the arithmetic and the spelling task, children indeed most often, but not exclusively, indicated that they thought they were correct in Grade 3 (Study 1; i.e., 93% of responses in arithmetic task, 77% in spelling task) and in Grade 2 (Study 2; i.e., 84% of responses in arithmetic task, 75% in spelling task). This result is in line with the high task performance Grade 3 (i.e., 94% accuracy in arithmetic task; 78% in spelling task) and in Grade 2 (i.e., 89% accuracy in arithmetic task; 70% in spelling task). Thus, children did indeed not just blindly select ‘correct’ on all metacognitive monitoring questions. This is also exemplified in the fact that average absolute metacognitive judgment (i.e., a judgment score of 3 when children indicated “Correct”, a score of 2 for “I don’t know” and a score of 1 for “Wrong”) were higher for correct academic answers than for incorrect academic answers in Grade 3 for arithmetic (Mcorrect = 2.95, SDcorrect = 0.07; Mincorrect = 2.24, SDincorrect = 0.62; t(124) = -12.73, p < .001, BF10 > 100) and spelling (Mcorrect = 2.80, SDcorrect = 0.20; Mincorrect = 2.47, SDincorrect = 0.32; t(145) = , p < .001, BF10 > 100); and in Grade 2 for arithmetic (Mcorrect = 2.86, SDcorrect = 0.20; Mincorrect = 2.22 , SDincorrect = 0.66; t(57) = -7.48, p < .001, BF10 > 100) and spelling (Mcorrect = 2.76, SDcorrect = 0.21 ; Mincorrect = 2.53, SDincorrect = 0.37 ; t(75) = , p < .001, BF10 > 100). These results demonstrate that children indicated higher confidence in their academic answer when that answer was indeed correct, and lower confidence when that answer was incorrect. As such, these results demonstrate that children did differentiate in their metacognitive judgments between correct and incorrect academic answers.

Reviewer 1, point 11

The authors controlled for IQ, and the scores on TTA and Dictation when examining the correlation between MMarith and MMspell in 8-9-year olds, but only controlled for IQ when examining the correlation between MMarith and MMspell in 7-8-year olds. While I understand that the correlation between MMarith and MMspell in 7-8-year olds was already not significant when only controlling for IQ, it may still be helpful to be more explicit and consistent on the different analytic approaches for the two age groups.

Author’s response: As the Reviewer indicated, we indeed did not control for scores on TTA and Dictation when examining the correlation between MMarith and MMspell in 7-8-year olds because this correlation was already not significant when only controlling for intellectual ability. We agree with the Reviewer that explicitly including this in the manuscript would increase transparency. Therefore, we added the following statement to the manuscript on page 25 : “Hence, further control analyses (i.e., in line with Study 1 in which the correlation between MMarith and MMspell was also controlled for performance on the TTA and Dictation) were not performed.”

Although we would contend that these analyses are not eligible to be included in the manuscript, to be fully transparent, we also calculated in the 7-8-year-olds the correlation between MMarith and MMspell with the scores on TTA and Dictation additionally included as control variables. As expected, metacognitive monitoring in one domain was not correlated to metacognitive monitoring in the other domain (r = .08, p = .54), nor were they predictive of each other (see Table R2).

 

Table R2. Regression analyses of MMarith and MMspell performance with metacognitive monitoring in the other domain and standardized task performance in both domains as predictors (Grade 2).

 MMarith

 � t p BFinclusion

Intellectual ability .35 3.34 .001 92.61

TTA .41 4.05 <.001 >100

Dictation .11 1.04 .30 1.14

MMspell .07 .62 .54 0.89

 MMspell

 � t p BFinclusion

Intellectual ability .21 1.57 .12 1.69

Dictation .32 2.58 .01 0.55

TTA -.05 -.34 .73 7.00

MMarith .09 .62 .54 0.72

Reviewer 1, point 12

“On one hand” should precede the phrase “On the other hand”. If the first hand is not present, the second hand is not really “the other hand”. The authors frequently use “on the other hand” without its preceding partner, “on one hand”. (e.g., page 4 line 8, page 5 line 16, page 27 line 1, page 28 line 15) Please review the paper and adjust the phrasing.

Author’s response: Thank you for drawing our attention to this phrasing issue. We have carefully adjusted this in the revised manuscript, such that “on the other hand” is always preceded by “on the one hand” or other indications of contrasting statements, such as “one possibility is that … . On the other hand, it is possible that …” (page 21). In paragraphs where this was not the case, other linking words were used (e.g., “furthermore”). All adjustments are indicated in colour throughout the manuscript.

Reviewer 1, point 13

The sentence “Materials consisted of standardized tests, paper-and-pencil tasks, and computer tasks…. (Page 9, Materials)” seemed to suggest that there were three types of tasks, and standardized tests were different from paper and pencil tasks. However, the standardized tests seemed to be the paper-and-pencil tasks. I would suggest rephrasing the sentence to “Materials consisted of standardized written tests and custom computer tasks….”

Author’s response: We fully agree with the Reviewer that this statement may confuse the reader. We have adjusted this sentence following the Reviewers suggestion (manuscript page 9): “Materials consisted of written standardized tests and computer tasks designed with Open Sesame [17].”

Reviewer 1, point 14

Although the authors stated that the computer tasks were arithmetic or spelling verification (page 9 line 8), the task descriptions suggested that the children were choosing the right answer (8+2 is 10 or 16) rather than verifying the answer (8+2=16, is the answer correct?). It would help avoid confusions by not characterizing them as verification tasks.

Author’s response: We thank the Reviewer for pointing to this ambiguity and apologize for being unclear. The arithmetic and spelling task should indeed not be categorized as verification tasks, as within these tasks, children had to select the correct of two response alternatives. The tasks were thus multiple choice tasks in nature. We have corrected this in the materials section on page 9 (i.e., “Namely, both tasks were multiple choice tasks with specifically selected age-appropriate items”) and we have removed the reference to verification tasks in the manuscript on page 9 (arithmetic task) and page 11 (spelling task).

Reviewer 1, point 15

In tables 2 and 3, some ts and ps are in uppercase. I think they should all be lowercase.

Author’s response: Thank you for highlighting this issue. We apologize for this mistake and have adjusted this in the manuscript.

Responses to Reviewer 2

The reported paper has many strengths, including the use of two samples of different ages, monitoring data in multiple domains (arithmetic and spelling), and two different skills tests in each domain (one on which monitoring was also measured and one that was standardized). The analyses answer a critical question in the literature and make a novel contribution. The design is technically sound, the writing is clear, and the claims appear supported by the data.

I have several comments for the authors to consider.

Reviewer 2, point 1

First, the conclusions will be better supported if the authors provide additional potential explanations for the different results across ages. The current discussion highlights how their results are consistent with previous work. For example, they write, “we are able to confirm the theoretically assumed development of metacognition from highly domain- and situation specific to more flexible and domain-general with practice and experience.” However, the current results suggest this may be tied to a fairly narrow time frame (between the ages of 7 and 9). Why does metacognition shift to being more domain-general? Why does this occur around ages 8 and 9? What kinds of practice and experience are theorized to be related to this shift? Additional insights into why this shift occurs around this age would help situate the novel empirical findings into the broader theoretical landscape of metacognition.

Author’s response: We agree with the Reviewer that adding potential explanations for the different results across ages would further our understanding of the development of metacognitive monitoring, yet we contend that such explanations are not so easy to make and are at best speculative. Based on the results of the current study, driving mechanisms for this gradual development from domain-specificity to domain-generality of metacognitive monitoring cannot be determined. In the existing literature, there is a lack of research that empirically investigates this issue. The current study provides a first step towards an understanding of the domain-specificity or –generality of metacognition by focusing on a narrow age range in which this development could occur, in related and highly relevant domains for children’s (academic) development. Future research should build on these results to reveal driving mechanisms behind this development towards domain-general metacognitive monitoring. Determining the developmental trajectory of whether and how metacognition generalizes across domains is crucial, not only form a theoretical perspective (i.e., as it sheds light on how metacognition develops throughout childhood and thus furthers our understanding of the functioning and cognitive architecture of metacognition), but also from a practical perspective, as determining, on the one hand, when metacognition becomes domain-general, and on the other hand, which conditions drive such a generalization, could have important influences on, for example, how metacognition is stimulated through educational practice.

As indicated below, we speculate that both cognition and education may play a role in the development from domain-specificity towards domain-generality of metacognitive monitoring. Therefore, we added the following extensive discussion on this issue in the manuscript on pages 30-31, and indicated these are essential areas for future research. We thank the Reviewer for pointing us to this possibility. 

“Although the driving mechanisms for the gradual development from domain-specificity to domain-generality of metacognitive monitoring cannot be determined on the basis of the current study, it is important to reflect on why metacognition shifts to being more domain-general around the ages 8-9. The existing literature offers some theoretical possibilities, albeit speculatively, that should be investigated in future research. 

The development from more domain-specificity of metacognitive monitoring towards more domain-generality in this age group is likely reflective of a gradual transition that occurs in the development of primary school children (e.g., [18]). In early stages of this development, children’s metacognitive monitoring might still be highly dependent on the (characteristics of the) specific stimuli, while over development, through experiences of failure and success, and with practice in assessing one’s performance as well as in (academic) tasks (such as arithmetic and spelling), monitoring might become more generic. These hypotheses and our results can be further interpreted within the dual-process framework of metacognition (e.g., [19–21]), which Geurten et al. [7] also used to interpret their findings. According to this dual-process framework of metacognition [19–21], metacognitive judgments can, on the one hand, be experience-based, i.e., based on fast and automatic inferences made from a variety of cues that reside from immediate feedback from the task and that are then heuristically used to guide decisions. As such, these metacognitive judgments are task-dependent and probably difficult to generalize across domains. On the other hand, metacognitive judgments can be information-based, i.e., based on conscious and deliberate inferences, in which various pieces of information retrieved from memory are consulted and weighted in order to reach an advised judgment. These conscious and effortful judgments are more likely to generalize to other domains. Taken together with the current results, this dual-processing model of metacognition may suggest that in second grade, children preferentially rely on automatic inferences when making judgments, while improvements of metacognitive abilities may enable children in third grade to rely more often on conscious and deliberate information-based processes. 

Another explanation for the gradual shift from domain-specificity to domain-generality of metacognition could be that this development might be associated with the development in other general cognitive functions, such as working memory capacity or intellectual ability. For example, Veenman and colleagues [22] found that metacognitive skills develop alongside, but not entirely as part of intellectual ability. Growth in these other, general cognitive functions might enable a shift from domain-specificity to domain-generality of metacognition. 

Finally, the development from domain-specificity towards domain-generality might also be driven by education, as teachers instruct children on assessing their own performance, which is at first very focussed on specific tasks. Over development, children might internalise this into a semantic network of their own abilities, which in turn might generalise to other tasks and thus become more general. 

It is essential to note that none of the above-mentioned hypotheses can be empirically evaluated within the current study. The focus of the current study was on whether a development toward domain-generality in metacognitive monitoring occurs in primary school children, in related academic domains, and, secondly when this occurs. The question on how, i.e., what mechanisms lie behind this, and why this is the case at this age, are important questions for future research.”

Reviewer 2, point 2

Second, I was a bit surprised that age was not featured in the analyses at all. For example, within each study, children’s ages spanned a full year (e.g., ranging from 8 years, 4 months to 9 years, 4 months in Study 1). It seems reasonable to investigate whether age is correlated with the other metrics (e.g., arithmetic skills, metacognitive monitoring) and potentially control for any shared variance across them related to age. Also, an interesting aspect of the studies is that there is an overlap in children’s ages across the studies, despite the children being in different grades. Specifically, it appears that some children in Study 1 and some children in Study 2 are between 8 years, 4 months and 8 years, 8 months. The authors may be able to provide additional insight into this metacognitive “shift” by potentially performing supplemental analyses on 8s vs. 9s in Study 1 and 7s vs. 8s in Study 2. I realize the authors pre-registered their analyses, which is 100% desirable and laudable, but also means any analyses with age would be considered exploratory or supplemental. I would encourage the authors to consider additional analyses with age in the models. At the very least, the authors should provide a justification in the paper for the reasons they opted not to include age in their tables and models.

Author’s response: We thank the reviewer for highlighting this concern. Following the Reviewers suggestion, Pearson correlation coefficients were calculated between age and academic and metacognitive performance measures in both grades (see table R3 below). The associations between age and the other metrics were not statistically significant, and Bayes factors were all below 0.43, consequently pointing to evidence for the null hypotheses of no association between age and the variables under investigation. In line with the lack of significant correlations with age, post-hoc defined partial correlations and regression models to control for shared variance across age and the other metrics (see Tables R4-R8) indicate that including age in the analyses does not change the interpretation of the current results. However, to be fully transparent to the reader, and as the Reviewer suggested, we added the following statement on this issue in the manuscript on page 16 and 23 for Study 1 and Study 2 respectively and added these post-hoc analyses including age in appendix.

“Although not originally pre-registered, we additionally re-calculated all analyses below with chronological age as an additional control variable. Considering chronological age within grade in the analyses reported below did not change the interpretation of the results (Appendix C).” (Study 1 - manuscript page 16; Study 2 – manuscript page 23).

It is true that there is a small overlap between the age range of Study 1 and Study 2, as the Reviewer indicated. It is, however, important to note that this overlap was only due to two children in Study 2 that were in the age range of Study 1. We therefore re-calculated all analyses with the exclusion of these two children, but excluding the children from the analyses did not change the results. It is also important to note that, while the age of these two children from Study 2 overlapped with the age of the children from Study 1, all children in Study 1 were third graders and all children in Study 2 were second graders. There was no overlap in grades between the studies.

The analyses to compare groups within each study (i.e., 8 vs 9-year-olds in Study 1; 7 vs 8-year-olds in Study 2) suggested by the Reviewer to further investigate the development from domain-specificity to domain-generality were indeed not preregistered. One possibility could be to present such analyses as exploratory, as suggested by the Reviewer. However, there are two reasons that prevented us from calculating these analyses. Firstly, dividing the samples within each study into two groups based on age (e.g., 7 vs 8 year olds in Study 2) results in very small groups (e.g., only twelve 8-year-olds in Study 2) that are not suitable for reliable statistical analyses. Also, we did a priori not specifically sample children to have an equal number of 7/8 or 8/9 year-olds in this study. Secondly, and as indicated above, age was not correlated – with Bayes Factors indicating evidence for the null hypothesis - with performance (i.e., academic performance and metacognitive skills) in each of the studies, indicating that it is rather grade than chronological age that is determining performance. We therefore decided to not split the samples of Study 1 and 2 into specific age groups. 

Table R3. Correlation analyses of age and academic and metacognitive performance measures in both grades.

 Study 1 - Grade 3 Study 2 - Grade 2

 Age Age

Arithmetic performance 

Custom task 

Accuracy 

r -.07 .15

p .44 .22

BF10 0.14 0.31

Response time 

r -.09 -.09

p .28 .46

BF10 0.19 0.19

Standardized task 

r .06 .18

p .46 .15

BF10 0.13 0.42

Spelling performance 

Custom task 

Accuracy .07 -.14

r .43 .26

p 0.15 0.28

BF10 

Response time 

r -.13 -.13

p .12 .29

BF10 0.36 0.25

Standardized task 

r .06 .01

p .46 .94

BF10 0.14 0.15

Metacognitive Monitoring 

Arithmetic 

r -.14 .07

p .11 .59

BF10 0.37 0.17

Spelling 

r -.01 -.13

p .87 .28

BF10 0.11 0.26

Table R4. Partial correlations of metacognitive monitoring and academic performance measures in 8-9-year-olds (Grade 3).

 Arithmetic Spelling

 Custom task – Accuracya Custom task - RT b Standardized task (TTA) a Custom task - Accuracya Custom task -RT b Standardized task (dictation) a

Metacognitive Monitoring 

Arithmetic 

r .86 -.05 .43 .53 .11 .35

p <.001 .53 <.001 <.001 .22 <.001

BF10 >100 0.13 >100 >100 0.23 >100

Spelling 

r .53 -.15 .38 .93 -.04 .71

p <.001 .09 <.001 <.001 .68 <.001

BF10 >100 0.45 >100 >100 0.12 >100

Note. All correlations are additionally controlled for age. 

a Controlled for intellectual ability.

b Controlled for intellectual ability and motor speed on the keyboard.

 

Table R5. Partial correlations of metacognitive monitoring and academic performance measures in 7-8-year-olds (Grade 2).

 Arithmetic Spelling

 Custom task – Accuracya Custom task - RT b Standardized task (TTA) a Custom task - Accuracya Custom task -RT b Standardized task (dictation) a

Metacognitive Monitoring 

Arithmetic 

r .80 .37 .46 .16 .08 .17

p <.001 .001 <.001 .23 .52 .18

BF10 >100 20.31 >100 0.32 0.20 0.38

Spelling 

r .06 .11 .11 .89 -.01 .36

p .66 .42 .39 <.001 .92 .003

BF10 0.17 0.22 0.22 >100 0.16 11.93

Note. All correlations are additionally controlled for age. 

a Controlled for intellectual ability.

b Controlled for intellectual ability and motor speed on the keyboard.

Table R6. Partial correlations of metacognitive monitoring measures.

 Study 1 – Grade 3 Study 2 – Grade 2

 Metacognitive monitoring Spelling Metacognitive monitoring Spelling

Metacognitive monitoring Arithmetic 

r .41 a .17 b

p <.001 .19

BF10 >100 0.37

Note. a Partial correlation controlled for intellectual ability, arithmetic and spelling performance on the standardized tasks and age; b Partial correlation controlled for intellectual ability and age.

Table R7. Regression analyses of MMarith and MMspell performance with metacognitive monitoring in the other domain, standardized task performance in both domains and age as predictors (Grade 3).

 MMarith

 β t p BFinclusion

Age -.12 -1.77 .08 2.04

Intellectual ability .14 1.91 .06 2.12

TTA .27 3.68 <.001 84.62

Dictation -.12 -1.18 .24 1.06

MMspell .49 4.93 <.001 >100

 MMspell

 β t p BFinclusion

Age .01 .21 .84 0.19

Intellectual ability .08 1.28 .20 0.36

Dictation .55 8.49 <.001 >100

TTA -.001 -.01 .99 0.19

MMarith .34 4.93 <.001 >100

 

Table R8. Regression analyses of arithmetic performance (i.e., arithmeticacc and TTA) and spelling performance (i.e., spellingacc and dictation) with metacognitive monitoring in the other domain, standardized task performance in the other domain and age as predictors (Grade 3).

 Arithmetic

 Arithmeticacc TTA

 β t p BFinclusion β t p BFinclusion

Age -.04 -.52 .61 .29 .04 .48 .63 0.42

MMspell .53 4.99 <.001 >100 .22 1.95 .05 3.19

Dictation -.07 -.64 .53 .30 .19 1.68 .10 1.83

 Spelling

 Spellingacc Dictation

 β t p BFinclusion β t p BFinclusion

Age .13 1.75 .08 1.24 .10 1.18 .24 1.14

MMarith .50 6.06 <.001 >100 .25 2.80 .006 10.16

TTA .09 1.15 .25 0.68 .23 2.63 .01 11.67

Reviewer 2, point 3

Third, I was also a bit surprised by the lack of attention to characterizing children’s metacognition more broadly at these ages. The authors provide basic descriptive statistics (means, standard deviation, and range) in the supplemental materials. In general, children’s metacognitive monitoring seems to be quite good, with average calibration scores around 1.4 to 1.8 (out of 2). However, additional information could help shed light on how children are performing on this task. For example, regardless of their performance, how often do children think that they are correct vs. how often do children select that they do not know? Similarly, are average calibration scores higher on correct responses or incorrect responses? When children are “uncalibrated,” is it more often because they are overconfident (thinking they are correct when actually wrong) or because they are underestimating their skills (thinking they are incorrect when actually right). Do these metrics vary by discipline? These findings would not change current conclusions about domain-specificity, but would provide additional contributions by better characterizing children’s metacognitive monitoring on these tasks.

Author’s response: 

We thank the Reviewer for the interest in the specific characterisation of the children’s metacognitive monitoring performance. We have provided the requested information below. Because these analyses were not preregistered, because the results of these analyses are in line with results on metacognitive monitoring reported in the existing literature (e.g., overconfidence in children of this age, see Destan & Roebers, 2015), because these analyses do not answer any of our original research questions, and, as the Reviewer indicated, because they do not change the conclusions of the current study, these results were not discussed in detail in the manuscript. In the remainder of this response, we provide the details that the Reviewer is asking for. We contend that adding this information to the manuscript would make it unnecessarily complicated and therefore decided not to include it. However, if the editor and Reviewer think that this is absolutely critical, we are willing to reconsider this, bearing in mind that these analyses were not preregistered and should be represented as such.

The additional results are as follows. General performance on the metacognitive monitoring task was indeed, as the Reviewer indicated, quite good, although there were both inter- and intra-individual differences in performance. As also outlined in our Response to Reviewer 1, point 10, regardless of performance, in both the arithmetic and the spelling task, children most often indicated that they thought they were correct in Grade 3 (Study 1; i.e., 93% of responses in arithmetic task, 77% in spelling task) and in Grade 2 (Study 2; i.e., 84% of responses in arithmetic task, 75% in spelling task). This result is in line with the high task performance Grade 3 (i.e., 94% accuracy in arithmetic task; 78% in spelling task) and in Grade 2 (i.e., 89% accuracy in arithmetic task; 70% in spelling task). Average absolute metacognitive judgment (i.e., a judgment score of 3 when children indicated “Correct”, a score of 2 for “I don’t know” and a score of 1 for “Wrong”) was higher for correct academic answers than for incorrect academic answers in Grade 3 for arithmetic (Mcorrect = 2.95, SDcorrect = 0.07; Mincorrect = 2.24, SDincorrect = 0.62; t(124) = -12.73, p < .001, BF10 > 100) and spelling (Mcorrect = 2.80, SDcorrect = 0.20; Mincorrect = 2.47, SDincorrect = 0.32; t(145) = , p < .001, BF10 > 100); and in Grade 2 for arithmetic (Mcorrect = 2.86, SDcorrect = 0.20; Mincorrect = 2.22 , SDincorrect = 0.66; t(57) = -7.48, p < .001, BF10 > 100) and spelling (Mcorrect = 2.76, SDcorrect = 0.21 ; Mincorrect = 2.53, SDincorrect = 0.37 ; t(75) = , p < .001, BF10 > 100). When children are “uncalibrated”, it is mostly, in line with the existing literature (e.g., Destan & Roebers, 2015), because they are overconfident in both Grade 3 (i.e., 90% of uncalibrated answers in arithmetic task; 92% in spelling task) and Grade 2 (i.e., 84% of uncalibrated answers in arithmetic task; 75% in spelling task).

Reviewer 2, point 4a

Fourth, on a very minor note, I found two pieces of the method section to be a bit confusing. First, when describing the procedure, the authors write, “The participants completed all tasks in the same order in an individual session, two sessions in small groups of eight children and a group session in the classroom.” I assume this means each child participated in four sessions. Is this because some tasks needed to be assessed one-on-one and other tasks did not? I think it would help to clarify which tasks were administered in which sessions and to clarify the timing of these sessions (e.g., after the individual session, how many days later was the small group session? What time of the school year were these sessions? Etc.). [Second, (see point 4b)].

Author’s response: We thank the Reviewer for bringing this need for further clarification to our attention. Following the Reviewers suggestion, we have rephrased the procedure section (see below) to include information on which tasks were administered in the different sessions and included information on the timing of the sessions. The tasks were indeed distributed over different sessions due to practical reasons and task characteristics (i.e. some tasks needed to be administered one-on-one and others did not). By doing so, we also aimed to minimize effects of fatigue due to long testing sessions.

“All participants participated in four test sessions, which took place at their own school during regular school hours, and all completed the tasks in the same order. In the context of a larger project, all children first participated in an individual session of which the data are not included in the current manuscript. Second, a session in small groups of eight children took place, including the computerized spelling task and motor speed task; Third, a second session in small groups took place, including the computerized arithmetic task and motor speed tasks; Fourth, in a group session in the classroom the standardized arithmetic and spelling tests and the test of intellectual ability were administered. Sessions were separated by one to three days on average; they were never adjacent.” (manuscript page 8-9).

Reviewer 2, point 4b

[Fourth, on a very minor note, I found two pieces of the method section to be a bit confusing. First, (see point 4a)] Second, the authors describe the computerized arithmetic and computerized spelling tasks as “verification” tasks. This made me assume that a single problem/word was presented and the child had to verify (click yes or no) as to whether it was correct. However, in reality, the task included two simultaneous presentations of a problem/word, one that was correct and one that was incorrect. The child had to select the correct one. This is super minor, but it might be more appropriate to call it a selection task or recognition task rather than a verification task for ease of interpretation.

Author’s response: We apologize for the incorrect naming of the custom arithmetic and spelling task. The arithmetic and spelling task should indeed not be categorized as verification tasks, as within these tasks, children had to select the correct of two response alternatives. As outlined in our response to point 14 by Reviewer 1, we have corrected this in the materials section on page 9 (i.e., “Namely, both tasks were multiple choice tasks with specifically selected age-appropriate items”) and we have removed the reference to verification tasks in the manuscript on page 9 (arithmetic task) and page 11 (spelling task). 

 

REFERENCES

1. Nelson TO, Narens L. Metamemory: A theoretical framework and new findings. Psychol Learn Motiv. 1990;26:125–73. 

2. Morsanyi K, Cheallaigh NN, Ackerman R. Mathematics anxiety and metacognitive processes: Proposal for a new line of inquiry. Psychol Top. 2019;28(1):147–69. 

3. Schraw G, Moshman D. Metacognitive Theories. Educ Psychol Rev. 1995;7(4):351–71. 

4. Schraw G, Crippen KJ, Hartley K. Promoting self-regulation in science education: Metacognition as part of a broader perspective on learning. Res Sci Educ. 2006;36:111–39. 

5. Flavell JH. Metacognition and cognitive monitoring: A new area of cognitive-developmental inquiry. Am Psychol. 1979;34(10):906–11. 

6. Flavell JH. Cognitive development: Children’s knowledge about the mind. Annu Rev Psychol. 1999;50:21–45. 

7. Geurten M, Meulemans T, Lemaire P. From domain-specific to domain-general? The developmental path of metacognition for strategy selection. Cogn Dev. 2018;48:62–81. 

8. Vo VA, Li R, Kornell N, Pouget A, Cantlon JF. Young children bet on their numerical skills: Metacognition in the numerical domain. Psychol Sci. 2014;25(9):1712–21. 

9. Andraszewicz S, Scheibehenne B, Rieskamp J, Grasman R, Verhagen J, Wagenmakers E-J. An introduction to Bayesian hypothesis testing for management research. J Manage. 2015;41(2):521–43. 

10. Fleming SM, Dolan RJ. The Neural Basis of Metacognitive Ability. In: Fleming SM, Frith CD, editors. The cognitive neuroscience of metacognition. Heidelberg: Springer-Verlag; 2014. p. 245–65. 

11. Chua EF, Rand-Giovannetti E, Schacter DL, Albert MS, Sperling RA. Dissociating confidence and accuracy: Functional magnetic resonance imaging shows origins of the subjective memory experience. J Cogn Neurosci. 2004;16(7):1131–42. 

12. Chua EF, Schacter DL, Rand-Giovannetti E, Sperling RA. Understanding metamemory: Neural correlates of the cognitive process and subjective level of confidence in recognition memory. Neuroimage. 2006;29(4):1150–60. 

13. Chua EF, Schacter DL, Sperling R. Neural correlates of metamemory. J Cogn Neurosci. 2009;21(9):1751–65. 

14. Simons JS, Peers P V, Mazuz YS, Berryhill ME, Olson IR. Dissociation between memory accuracy and memory confidence following bilateral parietal lesions. Cereb Cortex [Internet]. 2010 [cited 2019 Dec 13];20:479–85. Available from: https://academic.oup.com/cercor/article-abstract/20/2/479/311985

15. Davidson PSR, Anaki D, Ciaramelli E, Cohn M, Kim ASN, Murphy KJ, et al. Does lateral parietal cortex support episodic memory?. Evidence from focal lesion patients. Neuropsychologia. 2008;46:1743–55. 

16. Berryhill ME. Insights from neuropsychology: pinpointing the role of the posterior parietal cortex in episodic and working memory. Front Integr Neurosci. 2012;6. 

17. Mathôt S, Schreij D, Theeuwes J. OpenSesame: An open-source, graphical experiment builder for the social sciences. Behavior Research Methods. 2012. 

18. Schneider W. The development of metacognitive competences. In: Glatzeder BM, von Müller A, Goel V, editors. Towards a theory of thinking. Heidelberg: Springer-Verlag; 2010. p. 203–14. 

19. Koriat A, Levy-sadot R. Processes underlying metacognitive judgments: Information- based and experience-based monitoring of one’s own knowledge. In: Chaiken S, Trope Y, editors. Dual process theories in social psychology. New York: Guilford; 1999. p. 483–502. 

20. Koriat A. Metacognition and consciousness. In: Zelazo PD, Thompson E, editors. The Cambridge handbook of consciousness. Cambridge, UK: Cambridge University Press; 2007. p. 289–325. 

21. Koriat A, Ackerman R. Choice latency as a cue for children’s subjective confidence in the correctness of their answers. Dev Sci. 2010;13(3):441–53. 

22. Veenman MVJ, Spaans MA. Relation between intellectual and metacognitive skills: Age and task differences. Learn Individ Differ. 2005;15(2):159–76. 

23. Destan N, Roebers CM. What are the metacognitive costs of young children’s overconfidence? Metacognition Learn [Internet]. 2015;10:347–74. Available from: https://link.springer.com/content/pdf/10.1007%2Fs11409-014-9133-z.pdf

---

## [Decision Letter · Decision Letter 1]

10 Feb 2020

PONE-D-19-26504R1

Metacognition across domains: Is the association between arithmetic and metacognitive monitoring domain-specific?

PLOS ONE

Dear Mrs. Bellon,

Thank you for submitting your revised manuscript to PLOS ONE. I have sent your paper back to the original reviewers and have now received their comments. Both are positive about this version and recommend acceptance. I agree with them.  However, the reviewers also provide some further suggestions to clarify a few points. Therefore, I would like to give you the opportunity to address these comments in a minor revision before acceptance. 

We would appreciate receiving your revised manuscript by Mar 26 2020 11:59PM. To enhance the reproducibility of your results, we recommend that if applicable you deposit your laboratory protocols in protocols.io, where a protocol can be assigned its own identifier (DOI) such that it can be cited independently in the future. For instructions see: http://journals.plos.org/plosone/s/submission-guidelines#loc-laboratory-protocols

We look forward to receiving your revised manuscript.

Kind regards,

Jérôme Prado

Academic Editor

PLOS ONE

Reviewers' comments:

Reviewer #1: I really appreciate the thorough and thoughtful responses from the authors, the additional analyses to demonstrate the robustness of findings, and the authors making the study and the findings extremely transparent! I only have three very minor comments for the authors to consider.

Introduction

A very minor thing, I noticed that “Rinne & Mazzocco [6]” is referenced in text but perhaps “&” should be replaced with “and” (pages 6-7). I think “&” is usually used in parentheses citations (e.g., One recent study found (Rinne & Mazzocco, 2013)….), and “and” is usually used when the citation is a part of a sentence (e.g., Rinne and Mazzocco (2013) found…)

Study 1

It seems that there were four practice items to familiarize children with the metacognitive monitoring measure in the custom computerized tasks (page 13). However, based on the description of the custom computerized spelling task (page 11), there were six practice trials proceeded the block with 60 trials that included the metacognitive monitoring measure. Are the six practice trials in the computerized spelling task different from the four practice items described in the metacognitive monitoring section? I thought the practice items described in the spelling section (second block of computer task) and in the metacognitive monitoring section would be the same but the number of trials were different so perhaps I was wrong.

S4 Supplementary Materials

General metacognitive knowledge seems to have no association with arithmetic and spelling performances at all in study 1(page 2) and study 2 (page 6). I wonder whether children scored very high on this questionnaire, and their scores restricted the range for correlations. Perhaps it would be good to report means and standard deviations of general metacognitive knowledge for both age groups.

* The data are not yet available. The authors indicated that the data set will be available after manuscript publication.

Reviewer #2: I am recommending acceptance as the authors have done a nice job addressing my prior concerns.

However, I just wanted to point out a minor thing. This is not necessarily a critique but hopefully a helpful suggestion to ensure novice readers don't get tripped up on the opening paragraph.

The authors edited the second sentence of the paper to address a prior concern. By doing so, they have inadvertently made it quite long and difficult to parse. Given that this sentence introduces the main topic, it might be worth breaking it up into several key ideas. For example, the first key idea is that "learning from mistakes is a facet of metacognition." The second key idea is the definition of metacognition. And the third main point is the definitions of declarative vs. procedural metacognition. I would suggest using three brief sentences so that it's easier to follow. Also, I still found it a bit confusing that "declarative" metacognition is defined as "the ability to assess" cognition and metacognitive monitoring is also defined as an ability to make "self-assessments." Since monitoring is a part of procedural metacognition, it's confusing that its definition is so similar to declarative metacognition. Because this paper does not deal with declarative metacognition, my suggestion is to cut this reference to it and just focus on the procedural aspect. "One critical component of metacognition is procedural metacognition, which is the..." There are a few other long sentences in the paper that seem to contain multiple big ideas that might be worth editing. Also, throughout the paper I would suggest trying to be consistent as to whether the authors refer to the children's ages or grades. For the most part, the authors refer to ages, but occasionally only refer to them by grade level. Consistency is helpful.

I look forward to seeing the published version.

---

## [Author Response · Author response to Decision Letter 1]

12 Feb 2020

Response to Reviewers - PONE-D-19-26504

Metacognition across domains: Is the association between arithmetic and metacognitive monitoring domain-specific?

Dear Editor and Reviewers,

We thank you for your careful rereading of our revised manuscript and response letter, and for your positive evaluation of our manuscript. We appreciate the time and effort that you have dedicated to reviewing our work. We have taken your comments into account in our revision. Please find below our point-by-point response to your comments and queries. To ease the identification of changes to the text in the revised manuscript, we have highlighted all changes by using coloured text.

Responses to the Editor

Editor, general point

Thank you for submitting your revised manuscript to PLOS ONE. I have sent your paper back to the original reviewers and have now received their comments. Both are positive about this version and recommend acceptance. I agree with them. However, the reviewers also provide some further suggestions to clarify a few points. Therefore, I would like to give you the opportunity to address these comments in a minor revision before acceptance.

Author’s response: We thank the Editor and the Reviewers for their positive evaluation of the manuscript and their suggestions for some further clarifications. We agree that these were areas with room for improvement and have revised the manuscript in accordance with these comments. Specifically, we have rephrased longer sentences in our manuscript, splitting them into shorter, clearer sentences. We further addressed the confusion about the number of practice trials in the custom academic tasks. In the Supplementary materials section on the general metacognitive knowledge questionnaire, we added the descriptive statistics of this measure. Lastly, we have thoroughly checked the references to age and grade in the manuscript to ensure consistency. In the remainder of this response letter, we provide a point-by-point response to the points raised by the two Reviewers.

Responses to Reviewer 1

Reviewer 1, general point

I really appreciate the thorough and thoughtful responses from the authors, the additional analyses to demonstrate the robustness of findings, and the authors making the study and the findings extremely transparent! I only have three very minor comments for the authors to consider.

Author’s response: We thank the Reviewer for his/her thoughtful, positive feedback on our revised manuscript and response letter.

Reviewer 1, point 1

Introduction

A very minor thing, I noticed that “Rinne & Mazzocco [6]” is referenced in text but perhaps “&” should be replaced with “and” (pages 6-7). I think “&” is usually used in parentheses citations (e.g., One recent study found (Rinne & Mazzocco, 2013)….), and “and” is usually used when the citation is a part of a sentence (e.g., Rinne and Mazzocco (2013) found…). 

Author’s response: We thank the Reviewer for highlighting this. We have changed this on pages 6-7 as indicated by the Reviewer and additionally checked the entire manuscript for sentences which had the same issue and adjusted this accordingly, i.e. on page 19 (i.e., Arithmeticacc and TTA instead of Arithmeticacc & TTA). 

 

Reviewer 1, point 2

Study 1

It seems that there were four practice items to familiarize children with the metacognitive monitoring measure in the custom computerized tasks (page 13). However, based on the description of the custom computerized spelling task (page 11), there were six practice trials proceeded the block with 60 trials that included the metacognitive monitoring measure. Are the six practice trials in the computerized spelling task different from the four practice items described in the metacognitive monitoring section? I thought the practice items described in the spelling section (second block of computer task) and in the metacognitive monitoring section would be the same but the number of trials were different so perhaps I was wrong.

Author’s response: We fully agree with the Reviewer that that this wording may confuse the reader. The Reviewer is correct in assuming that the practice trials described in the introductory paragraph on the metacognitive monitoring task (page 13) are the same as those described in the individual sections on the custom tasks. In the custom spelling task, there were six practice items, in the custom arithmetic task, there were four practice items. The rationale behind the number of practice items was twofold. Firstly, in both the spelling and the arithmetic task, the number of practice items reflected the number of different task components. In the spelling task, three specific spelling rules were included in the task. The practice items thus encompassed two items per rule, yielding six items. In the arithmetic task, two operations were included (i.e., addition and multiplication). The practice items thus encompassed two items per operation, yielding four items. Secondly, the spelling task was the first task in the protocol for which children had to use the computer and in which the metacognitive protocol was implemented. To make sure children were familiar with these task requirements, the number of practice items was higher than in the arithmetic custom task, which was administered later in the protocol. Hence, when this arithmetic task was administered, children already had some experience with the general task requirements.

To prevent confusion about the number of practice items, we have revised the reference to practice items in the introductory paragraph on the metacognitive monitoring task (page 13), removing the reference to the number of trials. This informs the reader that practice items were administered, and the specific number of practice items for each task is then presented under its task description.

As a result, the manuscript includes the following references to the practice items:

- Manuscript page 13 (introductory paragraph on metacognitive monitoring task):

“To familiarize the children with the task, in each task practice items were presented.”

- Manuscript page … (paragraph on metacognitive monitoring in the arithmetic task):

“Each block was preceded by four practice trials to familiarize the child with the task requirements.”

- Manuscript page 11 (paragraph on metacognitive monitoring in the spelling task):

“Each block was preceded by six practice trials to familiarize the child with the task requirements.”

Reviewer 1, point 3

S4 Supplementary Materials

General metacognitive knowledge seems to have no association with arithmetic and spelling performances at all in study 1(page 2) and study 2 (page 6). I wonder whether children scored very high on this questionnaire, and their scores restricted the range for correlations. Perhaps it would be good to report means and standard deviations of general metacognitive knowledge for both age groups.

Author’s response: We completely agree with the Reviewer that reporting means and standard deviations of the general metacognitive knowledge questionnaire would improve the transparency of the findings on this task. We have added these for both age groups in the Supplementary material (i.e., S4). This resulted in the following addition to the supplementary materials:

S4 page 1 (i.e., Results Study 1): “The mean score on the general metacognitive knowledge questionnaire was 10.70 (SD = 2.35; range [4.00-15.00]).”

S4 page 6 (i.e., Results Study 2): “The mean score on the general metacognitive knowledge questionnaire was 6.65 (SD = 2.44; range [1.00-11.00]).”

 

Reviewer 1, point 4

The data are not yet available. The authors indicated that the data set will be available after manuscript publication

Author’s response: Indeed, we will make the data available through the Open Science Framework page of the current project after manuscript publication (https://osf.io/ypue4/?view_only=ce9f97af0e3149c28942a43499eafd32) .

 

Responses to Reviewer 2

Reviewer 2, point 1

I am recommending acceptance as the authors have done a nice job addressing my prior concerns. However, I just wanted to point out a minor thing. This is not necessarily a critique but hopefully a helpful suggestion to ensure novice readers don't get tripped up on the opening paragraph. The authors edited the second sentence of the paper to address a prior concern. By doing so, they have inadvertently made it quite long and difficult to parse. Given that this sentence introduces the main topic, it might be worth breaking it up into several key ideas. For example, the first key idea is that "learning from mistakes is a facet of metacognition." The second key idea is the definition of metacognition. And the third main point is the definitions of declarative vs. procedural metacognition. I would suggest using three brief sentences so that it's easier to follow. Also, I still found it a bit confusing that "declarative" metacognition is defined as "the ability to assess" cognition and metacognitive monitoring is also defined as an ability to make "self-assessments." Since monitoring is a part of procedural metacognition, it's confusing that its definition is so similar to declarative metacognition. Because this paper does not deal with declarative metacognition, my suggestion is to cut this reference to it and just focus on the procedural aspect. "One critical component of metacognition is procedural metacognition, which is the..." There are a few other long sentences in the paper that seem to contain multiple big ideas that might be worth editing. Also, throughout the paper I would suggest trying to be consistent as to whether the authors refer to the children's ages or grades. For the most part, the authors refer to ages, but occasionally only refer to them by grade level. Consistency is helpful.

I look forward to seeing the published version.

Author’s response: We thank the Reviewer for this constructive suggestion. We fully agree that due to the revision, this sentence has become quite long and difficult. We have altered the sentence as was suggested by the Reviewer, using shorter sentences instead of one long sentence (see below). 

We further agree with the Reviewer that the reference to declarative metacognition is not necessary for the current manuscript, as it does not deal with declarative metacognition. As such, it might cause confusion among the readers. As the Reviewer suggested, we have removed this reference to declarative metacognition, and adapted the sentence according to the Reviewers suggestion. This resulted in the following revised sentences in the manuscript on page 3:

“Learn from your mistakes” is an old saying that (grand)parents teach their children. This goes back to the premise that making mistakes is associated with learning. Noticing your mistakes is an example of monitoring your cognition. This monitoring of cognition is a facet of metacognition, a concept first introduced by Flavell [1]. One critical component of metacognition is procedural metacognition. This is a self-reflecting, higher-order cognitive process, which indicates how people monitor and control their cognition during ongoing cognitive processes [2,3]. Metacognitive monitoring is an important aspect of procedural metacognition and is defined as the subjective self-assessment of how well a cognitive task will be/is/has been performed [3,4].”

In addition to having revised the wording of the sentences pointed to by the Reviewer in the introduction, we have checked the entire manuscript for sentences that may be unnecessary long and/or containing multiple big ideas. We have adjusted the wording of these sentences to shorten them, without altering the content/message in them.

Lastly, we have checked the entire manuscript to ensure consistency in the reference to age and grade. Throughout the manuscript, we consistently used age, with additional reference (e.g., within brackets) to grade to make the distinction between the two samples as clear as possible. The reference to grade was maintained within the manuscript, because the educational grade of the children was important for the current findings, which indicated differences in results between grades, yet no association with chronological age. When specifically focussing on age within grade, we have used the term “chronological age within grade” to limit possible confusion among the readers.

 

REFERENCES

1. Flavell JH. Metacognition and cognitive monitoring: A new area of cognitive-developmental inquiry. Am Psychol. 1979;34(10):906–11. 

2. Flavell JH. Cognitive development: Children’s knowledge about the mind. Annu Rev Psychol. 1999;50:21–45. 

3. Nelson TO, Narens L. Metamemory: A theoretical framework and new findings. Psychol Learn Motiv. 1990;26:125–73. 

4. Morsanyi K, Cheallaigh NN, Ackerman R. Mathematics anxiety and metacognitive processes: Proposal for a new line of inquiry. Psychol Top. 2019;28(1):147–69.

---

## [Editor Report · Decision Letter 2]

19 Feb 2020

Metacognition across domains: Is the association between arithmetic and metacognitive monitoring domain-specific?

PONE-D-19-26504R2

Dear Dr. Bellon,

We are pleased to inform you that your manuscript has been judged scientifically suitable for publication and will be formally accepted for publication once it complies with all outstanding technical requirements.

With kind regards,

Jérôme Prado

Academic Editor

PLOS ONE

Additional Editor Comments (optional):

I noticed that the second sentence in the Introduction was duplicated. You may want to correct that before formal acceptance.

---

## [Editor Report · Acceptance letter]

21 Feb 2020

PONE-D-19-26504R2 

Metacognition across domains: Is the association between arithmetic and metacognitive monitoring domain-specific? 

Dear Dr. Bellon:

I am pleased to inform you that your manuscript has been deemed suitable for publication in PLOS ONE. Congratulations! Your manuscript is now with our production department. 

With kind regards,

on behalf of

Dr. Jérôme Prado 

Academic Editor

PLOS ONE